

# Synoptic drivers of co-occurring summertime ozone and PM2.5 pollution in eastern China

Lian Zong[1], Yuanjian Yang[1,*], Meng Gao[2], Hong Wang[1], Peng Wang[3], Hongliang Zhang[4], Linlin Wang[5], Guicai Ning[6], Chao Liu[1], Yubin Li[1], Zhiqiu Gao[1,5]

1. School of Atmospheric Physics, Nanjing University of Information Science & Technology, Nanjing, China

2. Department of Geography, Hong Kong Baptist University, Hong Kong SAR, China

3. Policy Research Center for Environment and Economy, Ministry of Ecology and Environment of the People's Republic of China, Beijing, China

4. Department of environmental science and engineering, Fudan University, China

5. State Key Laboratory of Atmospheric Boundary Layer Physics and Atmospheric Chemistry (LAPC), Institute of Atmospheric Physics, Chinese Academy of Sciences, Beijing, China

6. Institute of Environment, Energy and Sustainability, The Chinese University of Hong Kong, Shatin, N.T., Hong Kong, China

∗ Correspondence to: Dr./Prof. Y. Yang (yyj1985@nuist.edu.cn)





**Abstract**

In recent years, surface ozone ($O_3$) pollution during summertime (June-August) over eastern China has become more serious, and it is even the case that surface $O_3$ and $PM_{2.5}$ (particulate matter with aerodynamic diameter $\leq$ 2.5 μm in the air) pollution can co-occur. However, the synoptic circulation pattern related to this compound pollution remains unclear. In this study, the T-mode principal component analysis method is used to objectively classify four synoptic weather patterns

(SWPs) that occur over eastern China, based on the geopotential heights at 500 hPa during summertime from 2015 to 2018. Four SWPs of eastern China are closely related to the western Pacific subtropical high (WPSH), exhibiting, significant intraseasonal and interannual variations. Note that remarkable spatial and temporal disparities of surface $O_3$ and $PM_{2.5}$ pollution are given under these four different SWPs according to the ground-level air quality and meteorological

observations. In areas controlled by the WPSH or the prevailing westerlies, $O_3$ pollution is mainly caused by photochemical reactions of nitrogen oxides and volatile organic compounds under weather conditions of high temperature, moderate humidity and slight precipitation. In particular, the warm moist flow brought by the WPSH can promote hygroscopic growth of fine particulate matter in some local areas, resulting in the increase of $PM_{2.5}$ concentrations, which may form co-

occurring surface $O_3$ and $PM_{2.5}$ pollution. In addition, the low boundary layer height and frequency of light-wind days are closely related to the transmission and diffusion of pollutants under the different SWPs, modulating the levels of $O_3$-$PM_{2.5}$ compound pollution. Overall, our findings demonstrate the different roles played by synoptic weather patterns in driving regional surface $O_3$-$PM_{2.5}$ compound pollution, in addition to the large quantities of emissions, and may also provide

insights into the regional co-occurring high $PM_{2.5}$ and high $O_3$ level via the effects of certain meteorological factors.

**Keywords:** Synoptic weather pattern, ozone, $PM_{2.5}$, compound pollution, western Pacific subtropical high (WPSH)


## 1. Introduction

In recent years, China has been experiencing serious air pollution problems due to the



enormous emissions of polluting gases [e.g., sulfur dioxide, nitrogen dioxide ($NO_2$), etc.] and aerosol particulates [e.g., particulate matter with aerodynamic diameter ≤ 2.5 (10) μm in the air

$PM_{2.5}$ ($PM_{10}$), etc.]      associated with its rapid economic development, industrialization and urbanization, together with certain unfavorable meteorological conditions (Wang & Chen, 2016; Zhang et al., 2014; Y. Zhang et al., 2016). Particularly, atmospheric compound pollution has become serious (Li et al., 2019; Saikawa et al., 2017; C. Zhang et al., 2019), especially for the economically developed and densely populated eastern urban agglomerations of China, such as the Beijing–

Tianjin–Hebei (BTH), Yangtze River Delta (YRD) and Pearl River Delta (PRD) regions (Cai et al., 2017; Du et al., 2019; Ji et al., 2018; Li et al., 2020), exerting a severe threat in terms of public health, economy and society (Chen et al., 2019; Cohen et al., 2017; Day et al., 2017; Yim et al., 2019).

        In general, $PM_{2.5}$ pollution is featured with obvious diurnal and seasonal changes. Due to the

influence of atmospheric diffusion conditions such as precipitation and wind speed (WS), it tends to be enhanced in the morning and evening, lower at noon, and higher in winter and lower in summer (Amil et al., 2016; H. Liu et al., 2019; Ye et al., 2018; Zhang & Cao, 2015). The $PM_{2.5}$ level of China showed a steady increase from 2004 to 2007, and has since stabilized (Ma et al., 2016); however, there are still frequent $PM_{2.5}$ pollution events in autumn and winter (Song et al., 2017; Yang et al.,

2018; Ye et al., 2018; Zhang et al., 2014). In the past few years, the $PM_{2.5}$ concentration in China has decreased significantly as a result of measures introduced across the country that have reduced multi-pollutant emissions, adjusted energy structure, and increased supply of clean energy (Gui et al., 2019; Yang et al., 2020; Q. Zhang et al., 2019; Zhang et al., 2020). In contrast, summer $O_3$ pollution has gradually been prominent, replacing $PM_{2.5}$ as the primary pollutant in the air (Li et al.,

2019). As a secondary pollutant, ozone in the troposphere is mainly formed by photochemical reactions between $NO_x$, carbon monoxide (CO) and VOCs in the exposure of sunlight (Sillman, 2002). The prominent problem of $O_3$ pollution has attracted the attention of experts and scholars in recent years. For instance, Sun et al. (2016) showed that the observed summertime $O_3$ at Mt. Tai has increased significantly by 1.7 ppbv $yr^{-1}$ for June and 2.1 ppbv $yr^{-1}$ for July–August during the period

of 2003 to 2015. An increase of the maximum daily 8-h average concentration of $O_3$ (MDA8 $O_3$) at an annual-average rate of 4.6%, was reported by Fan et al. (2020), albeit with a decrease of the





frequency of PM$_{2.5}$ pollution.

Many studies have indicated that PM$_{2.5}$ and O$_3$ pollution are strongly correlated with local meteorological factors such as temperature, relative humidity (RH) and WS (Huang et al., 2016;

Miao et al., 2015; Shu et al., 2019; Tai et al., 2010). Miao et al. (2015) suggested that strong northwesterly synoptic winds, low BLH (boundary layer height), high RH and stable atmosphere are more prone to aerosol pollution in the BTH region during wintertime. Zhang et al. (2017) found that the majority of O$_3$ extremes occurred with daily maximum temperature (Tmax) between 300 K and 320 K, minimum RH (RHmin) less than 40%, and minimum WS less than 3 m s$^{-1}$, through the

analysis of extreme O$_3$ and PM$_{2.5}$ events from historical data (30 years for O$_3$ and 10 years for PM$_{2.5}$) in the US. Furthermore, the number of annual extreme PM$_{2.5}$ days was highly positively correlated with the extreme RHmin/Tmax days, and the correlation coefficient between PM$_{2.5}$ and RHmin (Tmax) was highest in urban and suburban (rural) regions. Shi et al. (2020) studied the spatial distribution of O$_3$-8h (O$_3$ 8-hour moving average) and PM$_{2.5}$, and their sensitivity of meteorological

parameters; pronounced positive (negative) correlation between temperature (BLH and absolute humidity) and O$_3$-8h was found, but the relation between WS and O$_3$-8h was spatially different; for PM$_{2.5}$, it was negatively (positively) correlated with temperature, WS and BLH (absolute humidity). Recently, Han et al. (2020) revealed that meteorological factors can explain ~46% of the daily variability in summertime surface O$_3$, while synoptic factors contribute to ~37% of the overall

meteorological effects on the daily variability of surface O$_3$ in eastern China. The abovementioned indicates that the variation of meteorological factors, which are mainly driven by the evolution of different weather circulation situations, play a non-negligible role in air pollution. Therefore, classification of air pollution according to the meteorological circulation has become particularly important, not least because of its worth when applied to air quality monitoring, forecasting and

evaluation (Liu et al., 2019; Ning et al., 2019; Yang et al., 2018; Zheng et al., 2015).

In recent years, it has become possible to objectively classify atmospheric circulation conditions using weather data such as GH, sea level pressure, WS and temperature, so that the weather mechanism of extreme weather can be better understood and analyzed. Compared with subjective weather classification, the objective approach has been widely used in air pollution

research (Beck & Philipp, 2010; Miao et al., 2017, 2019; Ning et al., 2018). Miao et al. (2019),



based on the daily 900 hPa GH fields during winter in Beijing, identified seven synoptic patterns using an objective approach, and found that the weak northwesterly prevailing winds and strong elevated thermal inversion layer, along with the local emissions of aerosols play a decisive role in the formation of heavy pollution in Beijing; noted also that the southerly prevailing winds can

transport the pollutants emitted from southern cities to Beijing. Zheng et al. (2015) studied the relationship between regional pollution and the patterns of large-scale atmospheric circulation over eastern China in October from 2001 to 2010 and identified six pollution types and three clean types. Specifically, weather patterns such as a uniform surface pressure field in eastern China or a steady straight westerly in the middle troposphere, particularly when at the rear of an anticyclone at 850

hPa, were found to be typically responsible for heavy pollution events. Many studies have suggested that $PM_{2.5}$ and $O_3$ pollution are mainly related to the East Asian summer monsoon (EASM) and western Pacific subtropical high (WPSH) (Li et al., 2018a; Xie et al., 2017; Yin et al., 2019; Zhao et al., 2010). The anomalous high-pressure system at 500 hPa, associated with downward dry, hot air and intense solar radiation can enhance the photochemical reactions to elevate the production of

tropospheric $O_3$ (Gong and Liao, 2019; Yin et al., 2019). Furthermore, Zhao & Wang (2017) and Yin et al. (2019) noted that the positive GH anomalies at high latitudes tend to significantly weaken the cold-air advection from the north and result in local high temperatures near the surface in northern China, while the WPSH can transport sufficient water vapor to the YRD region and lead to a decrease in surface $O_3$. In addition, different subregions can exhibit various distributions of

pollutant, even with identical emission scenarios (Li et al., 2019; Saikawa et al., 2017; C. Zhang et al., 2019). Also, it is still unclear how the distribution of pollution responds locally to large-scale atmospheric circulation patterns. Considering the reduced surface solar radiation due to $PM_{2.5}$ pollution (Haywood & Boucher, 2000; He & Wang, 2020; Huang et al., 2018; Yang et al., 2020) and thus suppressed the photochemical production of $O_3$ (Li et al., 2017), the compound $O_3$-$PM_{2.5}$

pollution-related meteorological conditions, should be complex and likely to be associated with certain weather types. Overall, the mechanism by which the synoptic weather pattern (SWP) modulates the characteristics of $O_3$-$PM_{2.5}$ compound pollution has yet to be comprehensively described.

In this study, the SWPs corresponding to the co-occurrence $O_3$-$PM_{2.5}$ pollution during





summertime are analyzed, focusing on the eastern China (108°–135°E, 17°–53°N). Then, the causes

of $O_3$-$PM_{2.5}$ compound pollution, as well as $O_3$-only pollution, from the perspective of the objective

classification of atmospheric circulation patterns, are revealed. The findings are expected to provide

a scientific reference for the monitoring, forecasting and evaluation of summertime air pollution in

eastern China.


## 2. Data and methods

The air quality data, including $PM_{2.5}$, $NO_2$, $O_3$, and $O_3$-8h, are from the national 24-h

continuous air quality observation published by the China Environmental Monitoring Station

(http://www.cnemc.cn/). The hourly observation data of a total from 949 stations (108°–135°E, 17°–

53°N) in eastern China during summertime of 2015–2018, which include the more prominent

pollution areas in the eastern urban agglomeration, such as the BTH, YRD, PRD, Guanzhong Plain

(GZP), Northeast Megalopolis (NEM) regions (the specific locations of stations and urban

agglomerations are shown in Fig. 1a). Surface meteorological data, such as Tmax, precipitation, WS

and RH from 517 meteorological observation stations and radiosonde data from 63 stations in

eastern China, were obtained from the China National Meteorological Information Center of the

China Meteorological Administration (http://data.cma.cn/site/index.html). The BLH was calculated

according to the method given by Guo et al. (2016, 2019), and the FLWD [frequency of light wind

(< 2 m s$^{-1}$) days], precipitation frequency (PF), and MDA8 $O_3$ were also counted.

Additionally, for synoptic analysis of particulate matter and $O_3$ pollution in summer, we use

the GH field at 500 hPa and wind field at 850 hPa from the NCEP/NCAR (National Centers for

Environmental Prediction/National Center for Atmospheric Research) daily reanalysis dataset on a

2.5° × 2.5° latitude/longitude grid during the study period.

The T-mode principal component analysis (T-PCA) is an objective mathematical computer-

based method that can be used to classify the synoptic circulation patterns of regional gridded data

in the troposphere at the lower level. Indeed, it is commonly regarded as the most promising weather

pattern classification method at present (Huth et al., 2008). Moreover, this approach has been widely

used in the studies of aerosols and $O_3$ pollution-related atmospheric circulation in China (Miao et

al., 2017, 2019; Ning et al., 2018, 2019). The T-PCA analysis module of the COST733 software



(http://cost733.met.no/) developed by the European Scientific and Technical Research Cooperation,

was used to classify the synoptic circulation pattern based on the 500 hPa GH field. More detailed

information about the T-PCA method can be found in Miao et al. (2017). To assess the performance

of synoptic classification and determine the number of classes, the explained cluster variance (ECV)

is selected in this study (Hoffmann & SchlüNzen, 2013; Ning et al., 2019; Philipp et al., 2014). The

detailed information about the ECV is provided in the supplementary document.

Based on the GB3095-2012 environmental air quality standard issued by the Ministry of

Ecology and Environment of China, $O_3$ ($PM_{2.5}$) pollution occurs when the MDA8 $O_3$ exceeds 160

(75) µg m$^{-3}$. To investigate the temporal variations of air pollution in various regions in summer,

the pollution levels of $O_3$ and $PM_{2.5}$ in each key area were verified according to their concentration

limits.

Finally, in order to make it clear in the analysis of different weather types of $O_3$ and $PM_{2.5}$

concentration change, we calculated the average distribution of $O_3$ and $PM_{2.5}$, as well as the

meteorological conditions for each type, and further calculated the anomalous distribution of these

variables, i.e., the average of $O_3$ and $PM_{2.5}$ and the average of the meteorological conditions under

the respective patterns minus the average during summertime of 2015–2018, were given as well.

The statistical significance was tested with a 0.05 confidence level via analysis of variance, which

enabled us to distinguish the significant differences of spatial distribution characteristics between

$O_3$ and $PM_{2.5}$ pollution under four SWPs.

### 3. Results

### 3.1 Spatial and temporal distribution of $O_3$ and $PM_{2.5}$ during summer 2015–2018

Figure 1 shows the summer averaged MDA8 $O_3$ and $PM_{2.5}$ concentrations at 949 stations in

the eastern region of China for 2015–2018. Among these stations, the MDA8 $O_3$ concentration at

most stations (662/949) exceeds 100 µg m$^{-3}$, of which 45 sites exceed 160 µg m$^{-3}$. The highest $O_3$

pollution is found in Zibo, Shandong, with a value of 181.5 µg m$^{-3}$. The averaged $PM_{2.5}$ at most

sites (680/949) is below 35 µg m$^{-3}$, while reaches 62.6 µg m$^{-3}$ in Handan, Hebei Province. On the

whole, the MDA8 $O_3$ and $PM_{2.5}$ in the BTH region and its surrounding areas is significantly higher

than in other regions; and besides, the level of $O_3$ in some urban clusters, such as the PRD, YRD,





GZP and NEM regions, is particularly higher than that of the surroundings, thus, we focus on analyzing these key areas later.

The temporal variations of $O_3$ pollution levels in key areas are displayed in Fig. 2, revealing an obvious increase in $O_3$ pollution levels and duration in the five priority areas for the period 2015–2018. The $O_3$ pollution in June is more severe than that in July and August, which is consistent with the fact that the peak concentration of MDA8 $O_3$ in northern China is in June (Gong and Liao, 2019). In spatial terms, $O_3$ pollution is particularly prominent in the BTH and GZP regions, with the number

of pollution days reaching 148 and 109, of which even 23 and 17 are moderate pollution, respectively. Figure 3 illustrates the $PM_{2.5}$ pollution-level variations and, although it is not difficult to see that $PM_{2.5}$ pollution weakens year by year, the number of days of $PM_{2.5}$ pollution in the BTH region reaches 192 (25 days for moderate pollution), which is higher than the number of days of $O_3$ pollution. The reduced visibility of haze days weakens the solar radiation reaching the ground and

inhibits photochemical reactions from generating $O_3$ (Li et al., 2019; Z. Zhang et al., 2015), as a result, the concentration of $O_3$ continues to increase with the mitigation of $PM_{2.5}$ pollution. It is worth noting that $O_3$ and $PM_{2.5}$ co-occurred in both the BTH (120 days) and GZP (60 days) regions. Overall, the $O_3$ and $PM_{2.5}$ concentration in eastern China exhibits distinct intraseasonal and interannual variations, indicating that, aside from the changes in emission sources (because it is

considered that inter-seasonal and short-term changes in emission sources are not significant), it may also be regulated by meteorological conditions, which is further analyzed below.

### 3.2 Objective classification of large-scale synoptic circulation patterns in summer

        To analyze the effect of meteorological conditions on the changes of $O_3$ and $PM_{2.5}$

concentration, it is necessary to statistically analyze the large-scale weather circulation situation in summer. Existing studies have shown that the WPSH (500 hPa GH field with obvious anticyclonic characteristics, and downward flow around the center) in summer prominently regulates the weather and climate of East Asia (Lu, 2002), owing to its various location, shape and intensity (Ding, 1994). A low-level southerly monsoon formed at the periphery of the WPSH can transport warm and humid

air from the ocean to East Asia, which might also be responsible for the asymmetric spatial distribution in response to an enhanced WPSH for ground-level $O_3$, i.e. a decrease in southern China



but an increase over northern China (Zhao & Wang, 2017).

Therefore, we used the T-PCA method to objectively classify the weather circulation of the 500-hPa GH field in the summers of 2015–2018, and finally obtained four SWPs related to the

movement and development of the WPSH. The westward extension and southward motion of the WPSH in Type 1, as shown in Fig. 4a, transports water vapor into the YRD region, and the prevailing southwesterly in the YRD region and westward flow from the north form a cyclonic convergence area, with high temperature and high humidity during the Meiyu season. For Type 2, it is noticed that the westerly trough deepens accompanying the northward (or southward) advance (retreat) of

WPSH, and the GH over northern China at 500 hPa is higher compared with Type 1 (Fig. 4b). The southerly wind blowing from the ocean to the continent lies in front of the bottom of the high pressure, affecting the southeastern region, while northern China is mainly controlled by the westerly trough, and the rain belt moves northwards to the east of the YRD region. Under Type 3, the WPSH shifts further north with a westward extension, and disintegrates a closed high-pressure

monomer along the eastern coast of China, while the main body of the WPSH remains on the ocean (Fig. 4c). In this case, the YRD region is completely controlled by the monomer of the WPSH, which implies that the rainy season in the YRD region ends in midsummer and the weather becomes hot and dry. At the same time, the rain belt moves gradually northwards to the BTH and NEM regions. According to Fig. 4d, the monomer of WPSH under Type 4 continues to extend westwards

and shift northwards, controlling northern China for a long time, and with persistent high temperatures and a heat wave occurring in most parts of the eastern China.

Fig. 5 presents the daily and annual variations of the SWPs in the summers of 2015–2018. The advance of the WPSH in eastern China occurs in June and July, while gradual withdrawal occurs mainly in August, so Type 1 mainly appears in June, while Type 2, Type 3 and Type 4 occurs mainly

in July and August. Consequently, Type 1, Type 2, Type 3 and Type 4 appear for 167, 117, 52 and 32 days, respectively, during the study period. Since the movement of the WPSH is often affected by the activities of the surrounding weather system (such as typhoons, the Tibetan high, etc.) (Ge et al., 2019; Liu & You, 2020; Shu et al., 2016), there may be a short southward retreat during the WPSH's advancement (e.g., around 10 August 2018) and a short northward advance during its

process of retreat (e.g., 21 and 29 August 2016). In general, it can be seen that the WPSH also shows



evident intra-seasonal and interannual changes, which will inevitably regulate the weather, climate and environment changes in eastern China.

### 3.3 O$_3$ and PM$_{2.5}$ pollution characteristics under four SWPs

**3.3.1 Spatial characteristics**

We calculated the averaged and anomalous spatial distributions of the MDA8 of O$_3$ and PM$_{2.5}$ under the four SWPs. The averaged MDA8 O$_3$ under the four SWPs can be seen in Fig. S2. The O$_3$ concentration is relatively high in the area north of the Yangtze River under Type 1, and the high values of the MDA8 O$_3$ are mainly concentrated in the North China Plain (NCP) region, with a total

of 94 stations surpassing 160 μg m$^{-3}$. Type 2 O$_3$ pollution is slightly weaker than that for Type 1, and the MDA8 O$_3$ at the 72 sites exceeds 160 μg m$^{-3}$. The O$_3$ high-value areas lie mainly in the NCP, GZP and YRD regions under Type 4, and there are 37 stations larger than 160 μg m$^{-3}$. Of the four SWPs, the lowest overall MDA8 O$_3$ occurs under Type 3, with only one site exceeding 160 μg m$^{-3}$. It is also found that the regions experiencing significant positive deviations of the MDA8 O$_3$

from the summer mean are as follows: the BTH, YRD and NEM regions under Type 1, the BTH and GZP regions under Type 2, the middle of the YRD and PRD regions under Type 3, and the YRD, GZP and PRD regions under Type 4 (Fig. 6).

Analogously, Fig. 7 shows the anomaly and significance of difference of PM$_{2.5}$ under the four weather types, presented as positive anomalies in the south of the BTH and YRD regions under Type

1, in the BTH, GZP and PRD regions under Type 2, and in the GZP and PRD regions under Type 4. Due to the obvious seasonal variations of PM$_{2.5}$ concentration (higher in winter and lower in summer) (H. Liu et al., 2019; Miao et al., 2015), no site exceeds 75 μg m$^{-3}$ for the averaged PM$_{2.5}$ concentration. Even so, the level of PM$_{2.5}$ in the BTH region is still significantly higher under the four types than that for the other urban agglomerations (Fig. S3).


### 3.3.2 Pollution pattern differences in key areas

Air pollution in eastern China is principally found in dense urban areas such as the BTH and YRD regions (Gui et al., 2019; Han et al., 2019), so we took the BTH, PRD, YRD, GZP and NEM regions in the eastern region as key areas, counted the average daily changes of O$_3$ and PM$_{2.5}$ in



each key region under different weather patterns (Fig. 8), and calculated the over-limit ratio in key

regions via the stations × days statistics (see Table 1). The diurnal variation of $O_3$ is more obvious,

peaking at about 15:00 (Beijing time), while contrasting diurnal variations of $PM_{2.5}$ are given for

different regions. According to Figure 8 and Table 1, the following characteristics can be identified

for different urban clusters: (1) in the BTH region, the $O_3$ pollution of Type 1 and Type 2 is relatively

serious, their over-standard rates reach 47.1% and 54.2%, and the $PM_{2.5}$ pollution rates of Type 2

and Type 1 reach 18.8% and 16.3%, respectively; (2) in the PRD region, the over-standard rates

variation of $O_3$ and $PM_{2.5}$ is equable; (3) in the YRD region, the $O_3$ pollution over-limit ratio presents

as Type 1 > Type 4 > Type 2 > Type 3, $PM_{2.5}$ pollution largely appears in Type1, and both $O_3$ and

$PM_{2.5}$ in Type 1 are higher than those in the other types; (4) in the GZP region, the $O_3$ pollution

frequency is higher in Type 2 and Type 4, and $PM_{2.5}$ pollution occurs more frequently in Type 2;

and (5) in the NEM region, $O_3$ pollution is always found in Type 1, Type 2 and Type 4, but the over-

standard rate is no more than 15% and $PM_{2.5}$ pollution in Type 1 is more than in Type 2.

In summary, Type 1 is prone to the formation of compound pollution of $O_3$-$PM_{2.5}$ (that is, when

the ground MDA8 $O_3$ concentration exceeds 160 μg m$^{-3}$, the $PM_{2.5}$ concentration also exceeds 75

μg m$^{-3}$) in the southern BTH and northern YRD regions, which can be denoted as "South BTH –

North YRD $O_3$-$PM_{2.5}$ compound pollution". Similarly, Type 2 can be denoted as "BTH – GZP $O_3$-

$PM_{2.5}$ compound pollution", Type 3 as "BTH – GZP $O_3$-only pollution", and Type 4 as "GZP $O_3$-

$PM_{2.5}$ compound pollution with BTH – YRD – PRD $O_3$-only pollution".

**4.  Discussion**

**4.1  Analysis of potential meteorological factors**

The activities of atmospheric circulation system often lead to changes in meteorological

elements, and to a large extent, affect the processes of pollutant formation, transmission and

diffusion.    Zhang et al. (2017) revealed that the extreme $O_3$ and $PM_{2.5}$ pollution events in the United

States always occur under the conditions of high temperature, low humidity and low WS, while

Miao et al. (2015) showed that RH is high when aerosol pollution occurs in the BTH region.

However, $O_3$ pollution in China is more frequent in summer, and the warm and humid flow brought

by the EASM makes summer always hot and moist. Zhao et al. (2019) investigated the RH of $O_3$



pollution in Shijiazhuang between 15 June and 14 July 2016, and found that the $O_3$ concentration

was higher at moderate humidity (RH average during daytime from 10:00 to 17:00 LT was 40%–50%).

Therefore, to explore the meteorological causes of $O_3$ and $PM_{2.5}$ pollution, we analyzed the distribution of the average and anomalies for Tmax, RH, PF, BLH and FLWD under the four SWPs (Figs. S4, S5, 9 and 10). Under the influence of the EASM, over 80% of the stations experience

high temperatures (Tmax > 27°C) in each SWP, although the anomaly of Tmax in Type 1 presents negative. Type 1 is characterized by humid condition in the south and dry condition in the north, with the rain belt mainly in the PRD and YRD regions; Type 2 has dry and wet anomalies meridionally in northern China, and the rain band is located in the middle of the BTH and YRD regions; the RH is large for most sites under Type 3 and Type 4, corresponding to the shifted rain

belt to the BTH and NEM regions under Type 3, and the heavy precipitation appeared in the western PRD region and the middle of the BTH and YRD regions under Type 4 (Fig. S4).

In terms of their anomalous spatial distribution, the positive anomalies of Tmax are located in the southern region of Type 3 and most of the eastern region of Type 4; and since Type 1 always appears in June, most areas are negative (Figs. 9a–d). For RH, Types 2, 3 and 4 are negative for the

south and positive for the north, while Type 1 is opposite (Figs. 9e–h). PF is characterized by positive anomalies in the area south of the Yangtze River under Type 1, in the YRD region under Type 2, in the BTH and NEM regions under Type 3, and in the area between the BTH and YRD regions under Type 4 (Figs. 9i–l). As can be seen from Fig. 10, when the BLH has a positive anomaly, on the contrary FLWD has a negative anomaly, which indicates the higher the height of the boundary layer,

the lower the frequency of light wind days, the more conducive to the diffusion of pollutants, and vice versa. After further inspection of Fig. S4, we found the YRD region in Type 1, the area north of the Yangtze River in Type 2, the BTH and PRD regions in Type 3, and most regions of Type 4 have shallow BLHs and high FLWDs, which is detrimental to the transportation of pollution in these areas, thus corresponding to high levels of pollution under these weather patterns.

**4.2  Effects of NO$_2$ on O$_3$**

Photochemical production of $O_3$ mainly involve emissions of VOCs and $NO_x$ from anthropogenic, biogenic and biomass burning sources (Deng et al., 2019; Gvozdić et al., 2011;



Sillman, 2002). Due to the implementation of a number of pollution abatement measures, the abundance of $NO_2$ has decreased significantly in eastern China in the past few years, leading to the

sensitive precursor of $O_3$ formation changing from VOCs to a mixture of VOCs and $NO_x$ (Wang et al., 2019b). As a result, both $NO_x$ and VOCs need being cooperatively controlled to mitigate $O_3$ pollution. Figure 11 shows the diurnal variations of $NO_2$ and the ratio of $O_3$ to $NO_2$ ($O_3$/$NO_2$) in five urban clusters, where the larger the $O_3$/$NO_2$ ratio, the more $O_3$ is generated by the photochemical reaction of $NO_2$. The $NO_2$ concentration reaches its lowest and $O_3$/$NO_2$ reaches a peak at around

15:00 (Beijing time) in the day, owing to the rapid consumption of $NO_2$ by the photochemical reaction under high temperature and strong solar radiation in the afternoon. As far as $O_3$/$NO_2$ ($NO_2$) is concerned, it can be seen that the daytime $O_3$/$NO_2$ ($NO_2$) values under Type 4 in the five regions is greater than (less than) under the other types, indicating that $O_3$ photochemical reactions are stronger under this type than the others. In contrast, the photochemical reaction of the BTH and

NEM regions under Type 3 and the YRD region under Type 1 is weaker owing to the warm and humid air brought by the WPSH, and the rainy weather is also unconducive to the occurrence of photochemical reactions to generate $O_3$. For the PRD region, the photochemical processes of the four SWPs are not significantly different. It can be seen that different weather patterns also have an important regulatory effect on the photochemical production of $O_3$.

In summary, the different SWPs can modulate the regional variability of summertime $O_3$ and $PM_{2.5}$ and their causes in summer as follows:

(1) Type 1: The area to the north of the Yangtze River under Type 1 is controlled by the westerly zone in the north of the WPSH at 500 hPa. Under the conditions of high temperature (Tmax > 27℃), moderate humidity (RH ~60%), and low PF, photochemical reactions are largely promoted

to cause severe $O_3$ pollution. Meanwhile, the area from south of the BTH region to north of the YRD region is located in front of the westerly trough, under the influence of the warm and humid air of the WPSH, and so the hygroscopic growth of fine particulates will cause a certain amount of $PM_{2.5}$ pollution, becoming $O_3$-$PM_{2.5}$ compound pollution (Fig. 12). In Figure S6, we counted the number and probability of occurrence of compound pollution days in each site in summer during 2015-2018,

indicating that high occurrence probability (maximum values can approach 46.7%) of compound pollution appeared in the Northern China plain (to the north of 32°N). About 55.6% of compound





pollution occurrence days at all sites occurred under Type 1.

(2)  Type 2: The westerly trough strengthens meridionally, and the northern region is still controlled by the westerly zone. Ozone pollution is severe under the meteorological conditions of high temperature, moderate humidity, and few precipitations. The $PM_{2.5}$ in the BTH region, which is located in front of the westerly trough, is high since the shallow boundary layer and low wind frequency are unfavorable for pollutant diffusion. Therefore, $O_3$-$PM_{2.5}$ compound pollution can also be rather frequent (Fig. 12). About 33.0% of compound pollution occurrence days at all sites occurred under Type 2 in summer during 2015-2018.

(3)  Type 3: High temperature, low humidity and few precipitations in the YRD region generates a large amount of $O_3$, while the positive BLH and negative FLWD anomalies are unfavorable to $O_3$ accumulation. On the other hand, summer typhoon activities will weaken the WPSH intensity over the YRD region, leading to the eastward retreat and northward shift of the WPSH, high WS in coastal areas that will ease $O_3$ pollution on the ground (Shu et al., 2016), and high PF in the BTH and PRD regions which tend to suppress $O_3$ production. Accordingly, Type 3 is characterized as light $O_3$-only pollution in the areas of the BTH, YRD and PRD (Fig. 12).

(4)  Type 4: High temperatures, medium-high humidity and few precipitations in the GZP and PRD regions can cause $O_3$-$PM_{2.5}$ compound pollution, but $PM_{2.5}$ pollution in the PRD region is not heavy, which is possibly in relation to local lower pollutant emissions. Under the control of the WPSH, there are strong photochemical reactions at high temperatures and little rainfall in the BTH region, which is also conducive to $O_3$ generation (Fig. 12). Meanwhile, relative to Type 1, $O_3$ pollution is lighter in the BTH, due to the differences of RH, BLH and FLWD.

## 5  Conclusions

In this study, T-PCA, an objective classification method, was applied to classify the 500-hPa weather circulation pattern as four SWPs in the summers of 2015–2018. It was found that these four SWPs are closely related to the development of the WPSH. The spatial and temporal distribution characteristics of $O_3$ and $PM_{2.5}$ pollution in eastern China under the four SWPs were analyzed to regulate and differentiate $O_3$ and $PM_{2.5}$ pollution in key areas. We find two synoptic patterns are prone to lead to co-occurrence of $O_3$ and $PM_{2.5}$ pollution: in the southern BTH and northern YRD regions under Type 1 and the areas in the BTH and GZP under Type 2 are associated with the double





high level of $O_3$ and $PM_{2.5}$. About 55.6% of compound pollution occurrence days at all sites occurred under Type 1 while 33.0% under Type 2.

Type 1 weather pattern appears frequently in June, with a stable WPSH ridge line at about 22°N, and the warm and humid air brought by the WPSH reaches the area south of the Yangtze

River, where a high temperature and high humidity Meiyu season is formed which suppresses the photochemical reaction of $O_3$ generation. Meanwhile, the north of China is controlled by a low-pressure trough with high temperatures and little rain. The hygroscopic growth of $PM_{2.5}$ occurs in the corresponding area in front of the trough with a small amount of water vapor transported by the WPSH, causing compound pollution of $O_3$ and $PM_{2.5}$ in the south of the BTH region and north of

the YRD region; particularly, unfavorable pollution diffusion conditions of a shallow BLH and low WS further exacerbate the severity of this compound pollution.

Under Type 2, the WPSH shifts northwards and retreats eastwards (sometimes retreats southwards and eastwards), with the meridional deepening of the East Asian major trough, and thus warm and humid airstreams are brought to the Northern China (e.g., the BTH region), gradually

elevating temperatures and humidity. Although positive RH anomaly promotes hygroscopic growth of the $PM_{2.5}$, water vapor leads a sink of ozone by contrast. As a result, the probability of double high level of $O_3$ and $PM_{2.5}$ under Type 2 is less than Type 1, the extent of compound pollution in Type 2 is also narrowed. On the other hand, weak precipitation, shallow boundary layer and low wind speed tend to create favorable conditions for pollution maintenance.

In general, the location of the WPSH is tightly associated with $O_3$ pollution in eastern China, and the changes of meteorological conditions in different regions affected by the WPSH can induce significant regional differences in $O_3$ and $PM_{2.5}$ pollution. More importantly, the effects of various large-scale weather circulation patterns on the $O_3$-$PM_{2.5}$ compound pollution and their corresponding physical and chemical processes, have been clarified, which has important scientific

reference value in summer air-quality forecasts, as well as assessment and policy-making services.

**Data availability**

Hourly $PM_{2.5}$, $NO_2$, $O_3$, and $O_3$-8h data is published by the China Environmental Monitoring Station (http://www.cnemc.cn/). Surface meteorological data, such as Tmax, precipitation, WS and RH,



radiosonde data can be obtained from the China National Meteorological Information Center of the
       China Meteorological Administration (http://data.cma.cn/site/index.html). The NCEP/NCAR daily
       reanalysis        dataset        can        be        download        from
       https://psl.noaa.gov/data/gridded/data.ncep.reanalysis.html.

**Author contributions**

L. Zong: Methodology, Data Curation, Formal Analysis, Writing- Original draft preparation, Results
       Discussion, Writing- Reviewing and Editing;Y. Yang: Conceptualization, Methodology, Formal
       Analysis, Results Discussion, Writing- Reviewing and Editing;M. Gao, H. Wang, P. Wang, L. Wang,
       H. Zhang, G. Ning, C. Liu, Y. Li, Z. Gao: Results Discussion, Comments, Writing- Reviewing and
       Editing

**Competing interests**

       The authors declare that they have no conflict of interests

**Acknowledgments**

       This study was jointly funded by supported by the National Key Research and Development
       Program of China (2018YFC1506502) and the National Natural Science Foundation of China
440   (41871029).

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




**Table 1. Over-limit ratio and concentration of MDA8 $O_3$ and $PM_{2.5}$ calculated via stations × days in key urban clusters under four SWPs**

| Urban cluster | Month | Type1 | | | | | Type2 | | | | | Type3 | | | | | Type4 | | | | |
|---|---|---|---|---|---|---|---|---|---|---|---|---|---|---|---|---|---|---|---|---|---|
| | | Stas × days | MDA8 $O_3$ OLR | MDA8 $O_3$ Con | $PM_{2.5}$ OLR | $PM_{2.5}$ Con | Stas × days | MDA8 $O_3$ OLR | MDA8 $O_3$ Con | $PM_{2.5}$ OLR | $PM_{2.5}$ Con | Stas × days | MDA8 $O_3$ OLR | MDA8 $O_3$ Con | $PM_{2.5}$ OLR | $PM_{2.5}$ Con | Stas × days | MDA $O_3$ OLR | MDA $O_3$ Con | $PM_{2.5}$ OLR | $PM_{2.5}$ Con |
| BTH | 6 | 6416 | 57.3% | 172.5 | 16.7% | 49.2 | 122 | 91.8% | 209.6 | 3.3% | 46.0 | 59 | 62.7% | 176.0 | 32.2% | 67.1 | 0 | 0 | 0 | 0 | 0 |
| | 7 | 1716 | 25.6% | 134.2 | 21.2% | 51.8 | 3681 | 54.2% | 165.6 | 22.5% | 56.9 | 1356 | 33.8% | 141.8 | 12.9% | 46.7 | 577 | 43.5% | 153.7 | 8.8% | 49.5 |
| | 8 | 1188 | 18.9% | 118.5 | 6.9% | 33.1 | 2805 | 35.3% | 143.1 | 14.7% | 44.0 | 1671 | 18.9% | 121.2 | 10.1% | 42.5 | 1268 | 31.9% | 144.3 | 6.0% | 39.7 |
| | 6~8 | 9320 | 46.6% | 158.6 | 16.3% | 47.6 | 365 | 54.2% | 169.8 | 18.8% | 51.2 | 3086 | 26.3% | 131.3 | 11.7% | 44.8 | 1845 | 35.5% | 147.3 | 6.9% | 42.8 |
| YRD | 6 | 19329 | 26.2% | 127.2 | 4.4% | 36.8 | 11098 | 14.3% | 107.0 | 2.2% | 44.5 | 181 | 12.7% | 102.4 | 0.6% | 32.0 | 0 | 0 | 0 | 0 | 0 |
| | 7 | 5207 | 24.5% | 119.8 | 5.3% | 38.6 | 8459 | 22.0% | 120.1 | 0.6% | 28.0 | 4135 | 14.5% | 112.6 | 0.1% | 24.0 | 1743 | 30.9% | 139.5 | 0.3% | 30.0 |
| | 8 | 3593 | 25.5% | 127.6 | 0.3% | 28.6 | 19922 | 18.3% | 113.8 | 3.0% | 33.3 | 4993 | 16.9% | 116.7 | 0.0% | 23.7 | 3817 | 14.7% | 114.8 | 0.1% | 25.1 |
| | 6~8 | 28129 | 25.8% | 125.9 | 4.0% | 36.1 | 13754 | 17.4% | 112.5 | 1.7% | 30.6 | 9309 | 15.8% | 114.6 | 0.1% | 24.0 | 5560 | 19.7% | 122.5 | 0.1% | 26.7 |
| PRD | 6 | 5327 | 5.2% | 80.1 | 0 | 16.8 | 101 | 31.7% | 146.9 | 0 | 24.9 | 48 | | 77.6 | 0 | 13.2 | 0 | 0 | 0 | 0 | 0 |
| | 7 | 1431 | 20.3% | 107.2 | 0.6% | 25.3 | 3076 | 7.6% | 81.8 | 0.1% | 19.1 | 1079 | 13.3% | 92.4 | 0 | 19.3 | 461 | 8.0% | 82.1 | 0 | 17.9 |
| | 8 | 977 | 25.4% | 122.5 | 1.1% | 27.9 | 2316 | 20.6% | 108.5 | 0.6% | 28.0 | 1376 | 15.5% | 103.4 | 0.3% | 24.0 | 993 | 17.9% | 111.4 | 0.1% | 25.1 |
| | 6~8 | 7735 | 10.5% | 90.5 | 0.2% | 19.8 | 5493 | 13.5% | 94.3 | 0.3% | 23.0 | 2503 | 14.3% | 98.2 | 0.2% | 21.8 | 1454 | 14.8% | 102.1 | 0.1% | 22.8 |
| GZP | 6 | 1549 | 42.9% | 147.6 | 0.4% | 33.2 | 38 | 52.6% | 165.3 | 0 | 30.4 | 19 | 26.3% | 150.8 | 0 | 34.7 | 0 | 0 | 0 | 0 | 0 |
| | 7 | 1879 | 40.7% | 145.0 | 1.9% | 33.7 | 1168 | 47.4% | 160.3 | 0 | 30.2 | 432 | 24.3% | 133.3 | 1.2% | 31.7 | 173 | 26.6% | 144.1 | 0 | 26.4 |
| | 8 | 537 | 28.5% | 134.0 | 1.6% | 31.6 | 850 | 36.8% | 146.3 | 1.6% | 34.2 | 530 | 28.9% | 143.1 | 1.9% | 37.2 | 394 | 38.6% | 154.9 | 1.3% | 39.9 |
| | 6~8 | 368 | 15.2% | 112.9 | 0.9% | 32.4 | 2056 | 43.1% | 154.6 | 2.1% | 35.9 | 981 | 26.8% | 138.9 | 1.5% | 34.7 | 567 | 34.9% | 151.6 | 0.9% | 35.8 |
| NEM | 6 | 13126 | 17.9% | 121.0 | 1.7% | 26.4 | 243 | 64.2% | 175.3 | 0.4% | 31.1 | 120 | 13.3% | 123.3 | 0.8% | 37.6 | 0 | 0 | 0 | 0 | 0 |
| | 7 | 3422 | 8.9% | 106.8 | 2.3% | 25.2 | 7338 | 17.0% | 120.7 | 1.3% | 28.1 | 2722 | 5.7% | 93.0 | 0.7% | 20.8 | 1141 | 15.1% | 122.4 | 0.2% | 24.3 |
| | 8 | 2341 | 4.4% | 92.6 | 1.0% | 20.1 | 5520 | 6.3% | 91.0 | 1.3% | 21.0 | 3282 | 5.6% | 98.3 | 0.9% | 23.2 | 2507 | 7.8% | 97.8 | 0.2% | 19.5 |
| | 6~8 | 18889 | 14.6% | 114.9 | 1.7% | 25.4 | 13101 | 13.4% | 109.2 | 1.3% | 25.2 | 6124 | 5.8% | 96.4 | 0.8% | 22.4 | 3648 | 10.1% | 105.5 | 0.2% | 21.0 |

Notes: stas × days, stations × days; OLR, Over-limit ratio; Con, Concentration ($\mu g\ m^{-3}$).

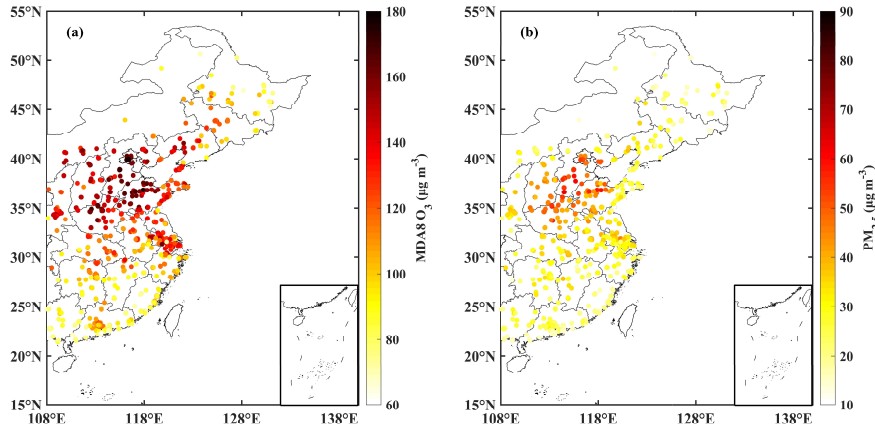

**Fig. 1. Average concentration of MDA8 O₃ (a) and PM₂.₅ (b) in eastern China during summers of 2015–2018. Stations and key urban clusters (red box) are shown in (a).**





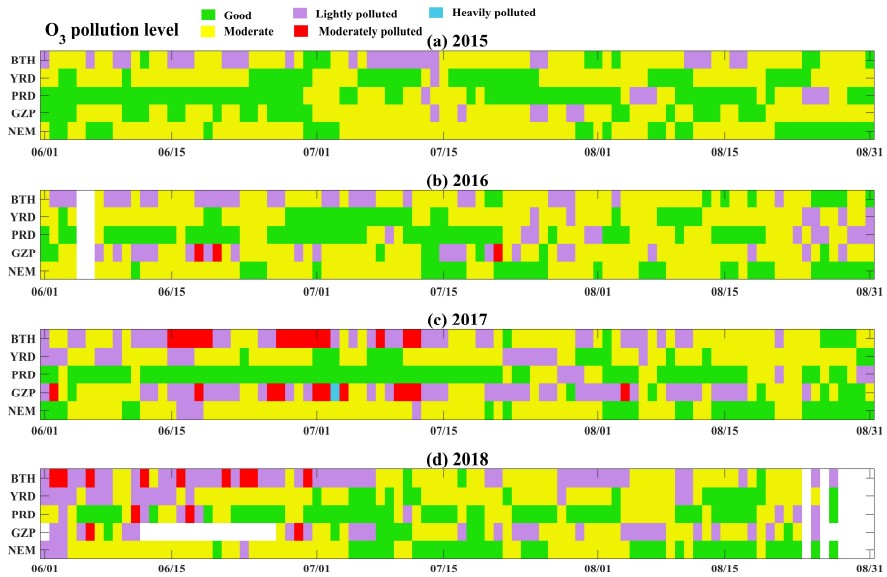


**Fig. 2. Time series of MDA8 O₃ pollution levels in key urban clusters.**



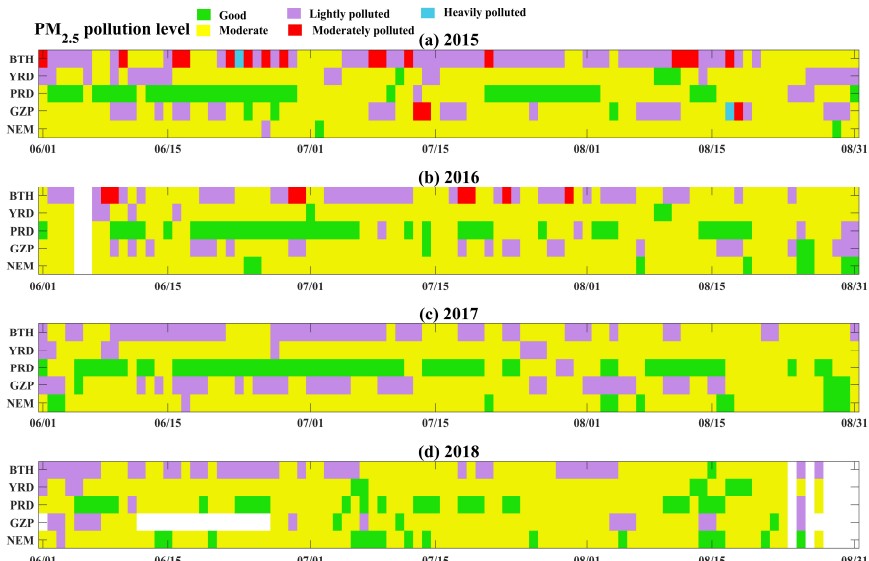

**Fig.3. Time series of PM$_{2.5}$ pollution levels in key urban clusters.**


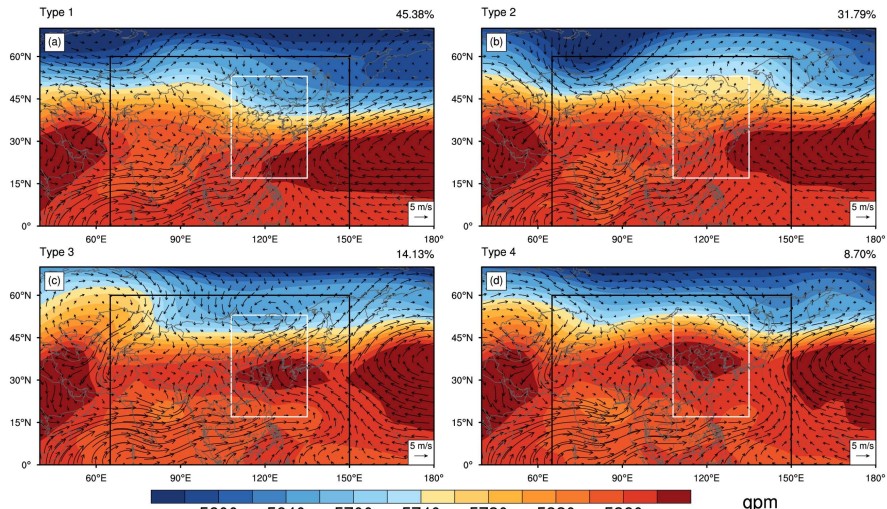

**Fig. 4. 850-hPa wind (vectors; see scale arrow at the bottom right in units of 5m/s) and 500-hPa GH (contours; see scale bar at bottom in units of gpm) patterns based on objective classification (see text for details). Black box area indicates the area for classification and the white box area is for the area of eastern China, the number at the upper right corner of each panel indicates the frequency of the occurrence of each pattern type.**






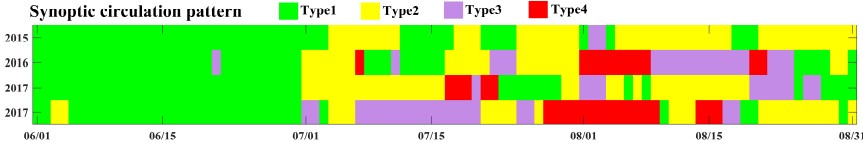

**Fig. 5. Time series of synoptic circulation pattern.**






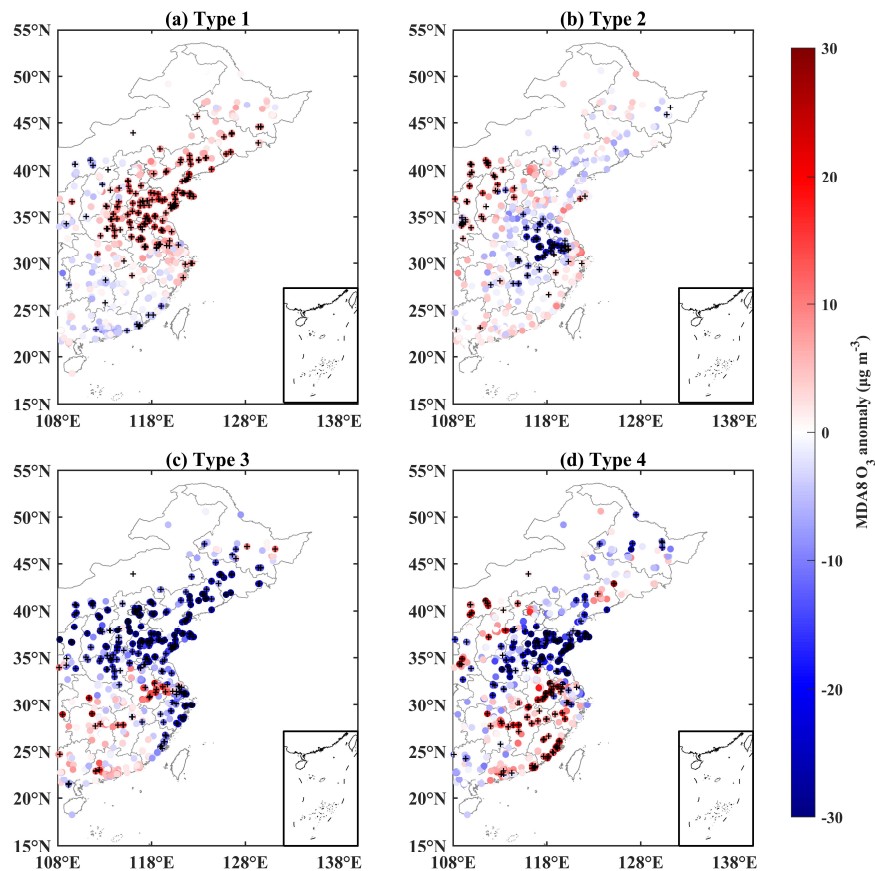

**Fig. 6. The MDA8 O₃ anomaly under four SWPs, where the sites marked with a '+' indicates**

**the Analysis of Variance passes the significance level of 0.05.**

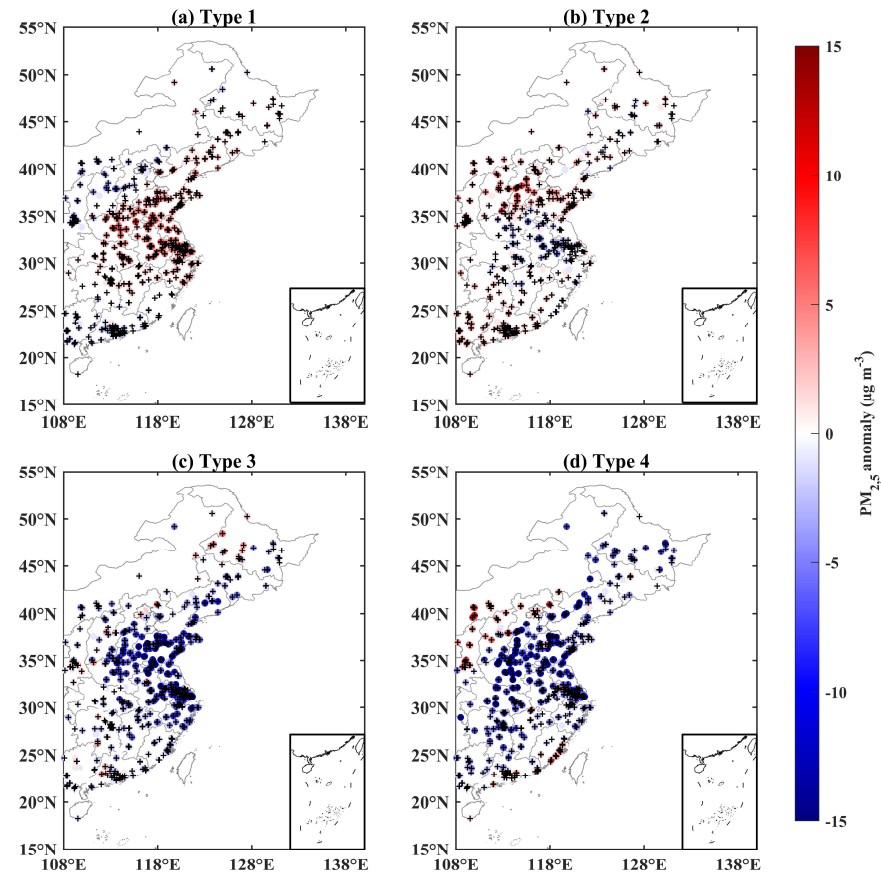


**Fig. 7. The PM$_{2.5}$ anomaly under four SWPs, where the sites marked with a '+' indicates the**

**Analysis of Variance passes the significance level of 0.05.**





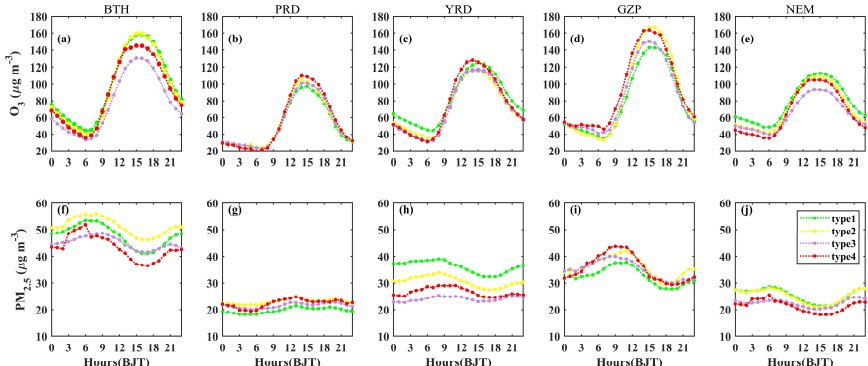

**Fig. 8. Daily variation of O₃ and PM₂.₅ under four SWPs in key urban clusters.**



**Fig. 9.** The anomaly of Tmax (a–d), RH (e–h), and PF (i–l) under four SWPs, where the sites

marked with a '+' indicates the Analysis of Variance passes the significance level of 0.05.


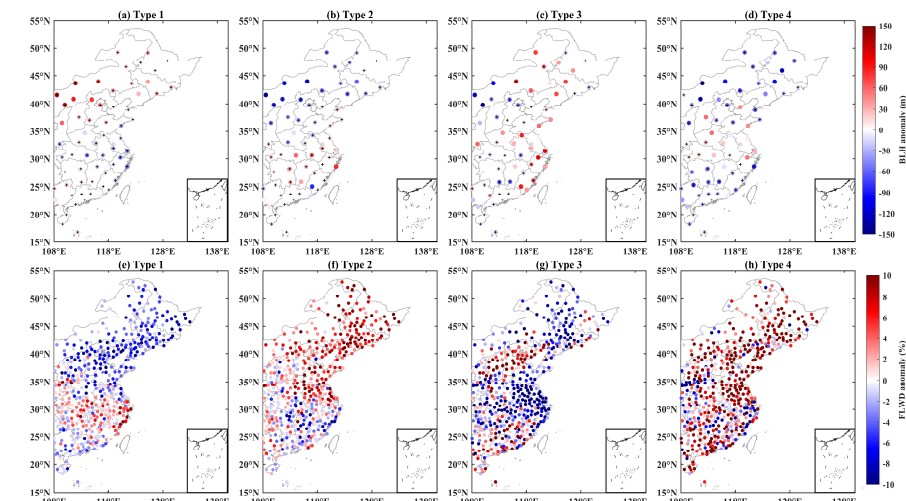

**Fig. 10. The anomaly and significance of difference of BLH (a–d) and FLWD (e–h) under four SWPs, where the sites marked with a '+' indicates the Analysis of Variance passes the significance level of 0.05.**





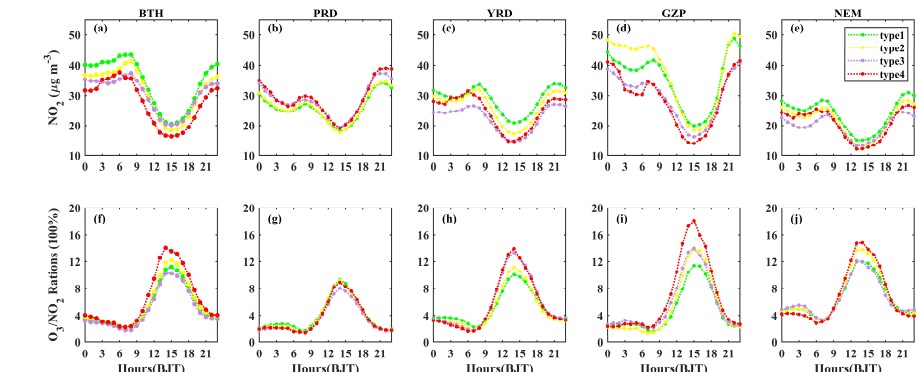


**Fig. 11. NO₂ and O₃/NO₂ daily variation under four SWPs in key urban clusters.**





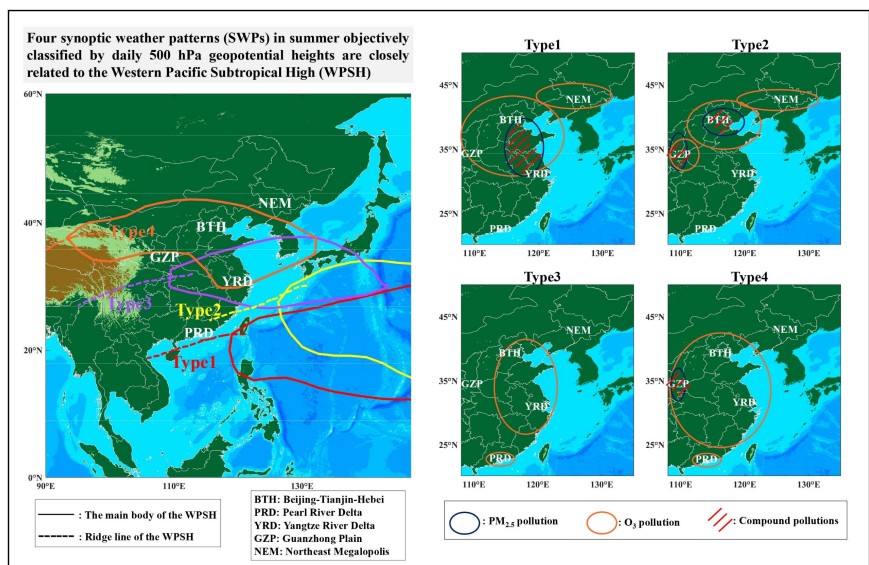

**Fig. 12. Schematic diagrams describing the relationships between the WPSH, four SWPs and**

725                **summertime O₃ and PM₂.₅ pollution in various regions.**