# Peer review of "Large-scale synoptic drivers of co-occurring summertime ozone and PM2.5 pollution in eastern China"

_Atmospheric Chemistry and Physics, 2020_

## Referee Comment (RC1) · Anonymous Referee #1 · 31 Aug 2020

It is appreciated that the authors took great efforts to elucidate the complex relationship between WPSH and air pollution in China during summer, which is interesting. However, the manuscript was not well organized (synoptic characteristics and their influences on pollution are better given in a section), and most analyses were superficial without observational evidence. The manuscript needs better organization and careful English editing.

1. Only four types were identified, which tends to oversimplify the complex synoptic situations during summer in eastern China. By contrast, six types were identified by Han et al. (2020) to understand the influence of synoptic weather on the summertime O3 pollution in eastern China; nine types were classified by Ye et al. (2016) for the aerosol pollution in the North China Plain. The region selected for the classification should be consistent with the studied region (eastern China in Fig. 1). Why use the region shown by the black squares in Fig. 4? And present it in a larger domain in Fig. 4? The classified region cannot resolve the processes needed? The selection of the classified region can significantly influence the classification results.

2. The classification results are odd. For example, in July of 2016, it quickly turned form Type 2 to Type 4, and then became Type 1 during a few days (Fig. 5). The WPSH cannot jump like that (e.g., from Type 2 to Type 4 in 24 hours). Besides, almost the whole June of 2016 was identified as Type 1 ("South BTH-North YRD O3-PM2.5 compound pollution"), while the South BTH and North YRD often

experience clean and pollution situations during a few days (https://www.aqistudy.cn/historydata/daydata.php?city=%E5%8D%97%E4%BA%AC&month=201606). The synoptic pattern of Type 1 cannot explain the formation and evolution of pollution in South BTH and North YRD. The variations of pollution level may be primarily controlled by other synoptic weather systems at 900/850-hPa, rather than the 500-hPa WPSH. The same problems also existed in other Types/regions. The classification results and their relationships with pollution are suspicious and unreliable.

3. Physically, the synoptic weather patterns influence the pollutions via PBL structure (e.g., large-scale subsidence), regional-scale transport of pollutants (e.g. PM2.5 and VOCs), and occurrence of precipitations. All these physical processes underlying were not well analyzed. The authors cannot just use simple correlation analysis to explain the mechanisms, which makes the conclusion unreliable and inconvincible. At least, typical pollution episodes of each synoptic weather type should be analyzed in-deep with observational evidence to validate the hypothesis given.

4. "The BLH was calculated according to the method given by Guo et al. (2016, 2019), and the FLWD [frequency of light wind (< 2 m s−1) days], precipitation frequency (PF), and MDA8 O3 were also counted." The detailed information about BLH should be directly given. At present, only a few cities have afternoon soundings during summer. Only 08:00 and 20:00 LT soundings were used to calculate the BLH, which

is inappropriate for this study. How to calculate the FLWD and PF? Why only choose these specific parameters? Are they significantly correlated to the pollution levels in all the studied regions (e.g., BTH, YRD, GZP)? How about the precipitation intensity and amount, and its lasting time? How about the wind directions and wind shear, and associated transport of pollutants (PM2.5 and VOCs)?

5. In the Results and Discussion sections, the "rain belt", "Meiyu", "rain band", "heavy precipitation" were simply analyzed with very few observational evidence. Since this study focuses on the movement/evolution of WPSH in summer (rainy season), more in-deep analysis on the links between precipitation and pollutions should be given with observational evidence and typical episodes.

6. Section 4.2 "Effects of NO2 on O3". The best proxy of photochemical reactivity is the ozone potential efficiency (OPE) but not the ratio of O3 to NO2. High photochemical reactivity probably appeared with high O3 and NO2 concentration but reasonably with a low O3/NO2 ratio. It is not accurate to take the O3/NO2 ratio as the judging criteria.

Specific comments:

Fig. 5, two "2017"?

"BTH, YRD, PRD, Guanzhong Plain (GZP), Northeast Megalopolis (NEM) regions",

the locations of these studied regions should be clearly described in manuscript and presented in the Figure.

Line 165-166, "More detailed information about the T-PCA method can be found in Miao et al. (2017)." The detailed information of the method should be directly given.

Some literatures were not properly cited. Please carefully check the citations of the whole manuscript. Some are given below. "In general, PM2.5 pollution is featured with obvious diurnal and seasonal changes. Due to the influence of atmospheric diffusion conditions such as precipitation and wind speed (WS), it tends to be enhanced in the morning and evening, lower at noon, and higher in winter and lower in summer." Dose the author mean that the diurnal variations of precipitation and wind speed modulate the pollution level? It is odd. How about the emission and PBL?

"Summer O3 pollution has gradually been prominent, replacing PM2.5 as the primary pollutant in the air…" Is it true? At present, PM2.5 still is the dominant pollutant in China.

"Miao et al. (2015) suggested that strong northwesterly synoptic winds, low BLH (boundary layer height), high RH and stable atmosphere are more prone to aerosol pollution in the BTH region during wintertime …" The strong northwesterly winds would favor the dispersion of pollutants in winter.

"Shi et al. (2020) studied the spatial distribution of O3-8h (O3 8-hour moving average) and PM2.5, and their sensitivity of meteorological parameters; pronounced positive (negative) correlation between temperature (BLH and absolute humidity) and O3-8h was found, but the relation between WS and O3-8h was spatially different; for PM2.5, it was negatively (positively) correlated with temperature, WS and BLH (absolute humidity)." It is another inappropriate citation. Please carefully read the previous study (Shi et al., 2020) and properly introduce it.

"Recently, Han et al. (2020) revealed that meteorological factors can explain ~46% of the daily variability in summertime surface O3, while synoptic factors contribute to ~37% of the overall meteorological effects on the daily variability of surface O3 in eastern China." More detailed information on Han et al. (2020) should be presented since it is quite similar to this study, such as its studied period, method, and classification results?

Line 95-97, "The abovementioned indicates that the variation of meteorological factors, which are mainly driven by the evolution of different weather circulation situations, play a non-negligible role in air pollution. Therefore, classification of air pollution according to the meteorological circulation has become particularly important …" The abovementioned literatures cannot support this statement.

Line 101-102, "In recent years, it has become possible to objectively classify atmospheric circulation conditions using weather data such as GH, sea level pressure…" It is not true. The objective classification method has been used since the 1990s.

Line 104-105, "the objective approach has been widely used in air pollution research (Beck & Philipp, 2010)…" Beck and Philipp (2010) didn't study the pollution issues.

Line 115-118, "Many studies have suggested that PM2.5 and O3 pollution are mainly related to the East Asian summer monsoon (EASM) and western Pacific subtropical high (WPSH) (Li et al., 2018a; Xie et al., 2017; Yin et al., 2019; Zhao et al., 2010)." More detailed information about these previous studies can be given, their studied periods, temporal scale and spatial scale. Seasonal variation? Inter-annual variation? The paper of Xie et al. (2017) has been withdrawn, please check (https://acp.copernicus.org/preprints/acp-2017-500/).

Line 129-131, "the compound O3-PM2.5 pollution-related meteorological conditions, should be complex and likely to be associated with certain weather types". This statement is not well supported.

Han et al. (2020) https://doi.org/10.5194/acp-20-203-2020

Ye et al. (2016) http://dx.doi.org/10.1016/j.atmosenv.2015.06.011

---

## Referee Comment (RC2) · Anonymous Referee #2 · 4 Sep 2020

This paper investigated the co-occurring of ozone and PM2.5 pollution of eastern China in summer. Four synoptic weather patterns (SWPs) were detected and the air pollution feature under each SWP was analyzed. The paper is not well organized and difficult to follow. The authors should revise the manuscript carefully to meet the standard of ACP. The detailed comments are listed below.

Major comments

This paper investigated the co-occurring of ozone and PM2.5 pollution in summer. However, the probability of PM2.5, ozone and compound pollution in each site could not be found in the manuscript. Please show their distribution at beginning of the manuscript. Their distribution under each SWP should be also shown to see the impacts of different weather patterns.

[Figure]

In this work, four synoptic weather patterns were detected and their difference were compared. The physical understanding the SWPs could be improved. In my opinion, Type 1 and type 2 represents normal WPSH pattern during early and late summer, respectively. Type 3 and type4 reflects two splitting states (southern mode and northern mode) of the WPSH, which mainly occurs in late summer. The climate background of early and late summer is quite different from each other. Thus, the SWPs of early and late summer should be compared separately.

The abstract is not well written and following information should be included: 1) The distinct feature of each synoptic weather pattern; 2) the pollution features under each SWP; 3) which SWP mostly favors the co-occurring of ozone and PM2.5 pollution; 4) where the co-occurring of ozone and PM2.5 pollution happens.

Other comments

Line 26: Please describe details of SWPs here.

Line 30: How about the ozone pollution over the regions where are not controlled by the WPSH or the prevailing westerlies. Where are these regions?

Line 34: Please explain the meaning of "some local areas".

Line 35: Please clarify where the co-occurring surface O3 and PM2.5 pollution happens.

Line 35-36: How the WPSH affects the boundary layer height and frequency of light-wind days?

Line 36: What does the "different roles" mean? Please provide some explanations.

Line 118-120: In abstract, it is mentioned that high temperature, moderate humidity and slight precipitation favors the ozone formation. It is not consistent with the statements here.

Line 124: It is not consistent with the conclusions in abstract. In abstract, it is found

that the warm moist flow brought by the WPSH result in co-occurring surface O3 and PM2.5 pollution.

Line 148: I cannot find the position of the urban agglomeration in Figure 1a. Please check.

Line 207: Please show the temporal variations of co-occurring events.

Line 229: Please focus on the position and strength of WPSH.

Line 229: It should be the north advance of WPSH.

Line 234: Type 1 and type 2 represents normal WPSH pattern during early and late summer, respectively. Type 3 and type4 reflects splitting of the WPSH, which mainly occurs in late summer.

Line 260: Type1 mainly occurs in June and it explains why the O3 concentration is higher.

Line 295: Please show the spatial distribution of the co-occurring surface O3 and PM2.5 pollution under each SWP.

Line 300: Potential meteorological factors should be included in the results of this paper.

Line 305-310: This part should be mentioned in introduction.

Line 315-316: It is because the type 1 mainly occurs in early summer.

Line 365: Figure S6 shows probability of occurrence of compound pollution days under all four types. Please show the situation under type1.

Figures and tables:

Figure 2 and 3: Please Mark the heavy polluted cases with dark color.

Figure 4: Please draw the ridgeline of the subtropical high.

Figure 8 and Figure 11: Please compare the difference of daily mean values. An introduction of daily variations makes things difficult to follow.

---

## Referee Comment (RC3) · Anonymous Referee #3 · 10 Sep 2020

Review of acp-2020-596: Synoptic drivers of co-occurring summertime ozone and PM2.5 pollution in eastern China

The authors use the T-mode PCA to objectively classify the summertime synoptic weather pattern across East-Asia and the western Pacific Basin aiming to identify the mode(s) most favorable for compound pollution events across sub-regions in China, specifically for PM2.5 and O3. Many factors governing these events operating across an array of scales are explored. The PCA identified 4 synoptic regimes characterizing the seasonal set up of the 500 hPa WPSH from 2015-2018. An additional large-scale circulation is also at work here, the East-Asian monsoon, which is discussed in context to the WPSH. Additionally, the authors discuss the effects of precipitation frequency and boundary layer characteristics on regulating compound pollution events. Occur-

rences of pollution are based on Chinese governmental standards.

While this work has great upward potential to be a significant contribution to the community, many revisions are required before publication. Secs. 1-2 are written quite well and motivate the questions at hand. Beyond that however, I believe that more concrete connections can be and must be made between processes unfolding at different scales (synoptic down to the mesoscale) of motion that lead to Types 1 and 2 dominating the regulation of compound pollution events. For instance, connecting the modulations in the WPSH to changes in favorable PBL conditions and thermal stratification need to be made, in addition to changing precipitation amounts between the types. All of these processes dictate the amount of pollution in the atmosphere at any given time. The final sentence of Sec. 1 states that this manuscript will examine the SWPs responsible for co-occurring pollution events, but the synoptic scale processes have bearing on finer scale processes such as PBL characteristics that are critical for air quality (e.g. inversions associated with tropospheric sinking motion). The authors analyze changes in PBL height between the types and provide loose discussion of their implications for air quality, but further analysis is needed.

Major comments

1. The abstract needs to be shorted and be more specific. 2. How do the percentages of the PCs sum to 100%? Shouldn't there be other relevant synoptic patterns than just those 4, meaning that the leading 4 patterns account for most of the synoptic-scale pattern but not all? 3. The language used to describe the synoptic scale features needs to be presented in a manner consistent with meteorological standards (see Bluestein 1992). In its present form, it is very difficult to follow the discussion. Here is n example. On lines 226-227, the authors state "The westward extension and southward motion of the WPSH in Type 1, as shown in Fig. 4a, transports water vapor into the YRD region, and the prevailing southwesterly in the YRD region and westward flow from the north form a cyclonic convergence area, with high temperature and high humidity during the Meiyu season." The 850 hPa flow associated with each PC correlates highly with the

gradient in 500 hPa GH as rather expected, but what is meant by "southward motion of the WPSH?" Are the authors referring to the anticyclonic flow about the WPSH (i.e. northerlies to the west of the GH maximum)? Also, the sign of the relative vorticity should differ with height in the troposphere. For instance, should vorticity be negative in the lower troposphere (i.e. anticyclonic), it should be positive (i.e. cyclonic) in the upper troposphere (assuming a thermally direct circulation on a rotating sphere). Are the authors referring to the cyclonic shear vorticity anomaly apparent in the 850 hPa arrows around 120E/30N? The authors should use GH anomalies as reference points to describe the flow patterns, and they should make sure that it is clear which level in the atmosphere is being referenced in the text. More examples are given below. 4. Sec. 3.2: I feel as though the discussion of the PCs could be tied more explicitly to the vertical motion field. Obviously, the WPSH is characterized by mid-tropospheric downward vertical motion and doesn't need much justification. However, the strength of the sinking motion and its co-occurrence with low wind events is driven by the synoptic pattern and could be shown. I would suggest at least a supplemental figure showcasing how the vertical motion varies with PC, perhaps overlaid with the 10-m windspeed. This would set up the next section nicely, which returns to examining the spatial characteristics of PM2.5 and O3. 5. Diffusion of pollutants between the PBL and the free atmosphere is fundamentally related to the turbulent mixing and thermal stratification of the overlying atmosphere. While referenced here, I believe that this is an integral component of this work and must be explicitly addressed across the various subregions. How do the vertical profiles of temperature, moisture, and wind compare across the multiple PCs and subregions? How are these anomalies physically related to the different synoptic weather pattern differences between the PCs?

Other comments

1. Line 32: "Slight" should be "low" 2. Line 57: insert a "the" before "economy" 3. Line 72: Change "attached" to "caught" 4. Line 85: "US" should be "United States" 5. Line 105: The Miao et al. findings should be reproducible here but for a multitude of areas.

Cross-sections similar to their Figs. 6-7 would work, but regionally averaged vertical profiles would work as well. Vertical profiles of state variables (temperature, stability, vertical velocity, etc.) should be included in this manuscript as these quantities' vertical variation help to significantly modulate PBL and free atmosphere exchange of heat, moisture, pollution, etc. I would also suspect that summertime surface winds would be lower due to more infrequent midlatitude cyclone occurrences, so pollution "pooling" would be more frequent. 6. Lines 146-147: Subregions should be labeled in a figure to orient the reader. This can be done in panel (a) of Fig. 1. 7. Fig 1.: There is no "red box"? If there is, it is not clear 8. Figs. 2 and 3: Please change the color of "heavily polluted" regions to something other than turquoise. It can easily be misinterpreted as a "good" category 9. Line 200: How is "pollution day defined" for O3? By the thresholds laid out earlier (160 ug/m3 threshold)? Also, what constitutes "moderate" pollution? Same question applies for PM2.5. 10. Line 219-220: This low-level transport feature and its variation with PC is not shown in any figure but is referenced. I believe that at least one figure should show this feature since it is being discussed in forthcoming results 11. Line 226: How can you infer that water vapor is being transported into the YRD regions? The 850-hPa flow vectors are at best directed parallel to the YRD coastline. Otherwise they are directed offshore. Additionally, inferences about moisture transport should be made by wind/water vapor overlays or by integrated vapor transport/moisture convergence analysis (see Rahimi et al. 2018), which this figure does not have. 12. Line 227: Flow shifting from southwesterly to westerly with northward extent is anticyclonic, which we see in the figure. At the same time, we see a cyclonic sped shear maximum, so it is impossible without quantifying the vorticity explicitly to say if this is anticyclonic or cyclonic. Please remove "cyclonic." 13. Line 229: Is it the WPSH retreating or advancing? Its axis appears to shift north slightly... 14. Line 230: Consider deleting, "and the GH over northern China at 500 hPa is higher compared with Type 1 (Fig. 4b)." The change in the magnitudes of GH are not terribly important; it is the change in their gradients that regulates the winds in each PC. Line 231: The only Type 2 onshore flow (at 850 hPa) I see is around 120E

by 30N, which lies directly west of the Type 2 GH maximum. This is an example of how you can use certain language relating flow properties to GH anomalies for specific PCs. Currently the authors state, ". . .southerly wind blowing from the ocean to the continent lies in front of the bottom of the high pressure,. . .", which is very unclear. More generally, I recommend the authors refrain from using "top" and "bottom" unless they are referring to the vertical axis (i.e. up and down). 15. Line 233: "and the rain belt moves northwards to the east of the YRD region." Are the authors referring to the belt as it compares to other PCs? If so, the different PCs may be compared to one another, but it is not guaranteed that any type will necessarily evolve from another type. Please clarify and reword throughout the text. 16. Lines 237-238: ". . .which implies that the rainy season in the YRD region ends in midsummer and the weather becomes hot and dry." How is this implied? 850-hPa flow is onshore beneath the western edge of the 500 hPa monomer of the WPSH. This linkage is not implicit and should be explained. Moreover, references made to shifts in precipitation need to be explicitly shown if they are going to be frequently referred in the text. 17. Lines 239-240: "continues to extend westwards and shift northwards," shifts westward and northward compared to PC3. Again, please indicate it's the synoptic pattern's position more explicitly. Something like, "In Fig. 4d, the monomer is located north and west of the feature in Fig. 4c". The word accordingly relates this sentence to the previous one, but it isn't clear what that linkage actually is. Also, please explain the linkage or remove the word "accordingly." 18. Line 241: Heat wave? How is PC4 related to a heat wave? Where is this shown in the figures or analyses? 19. Figs 3-4: How are levels of air quality defined? Are they arbitrary? If so, then a brief justification is required. If they are a community standard, then a source is needed. 20. Fig. 4 shows the PCs of the synoptic weather pattern and associated percentages of occurrence for the study period. 21. Fig. 5: 2017 is labeled twice. Should the second instance be 2018? 22. Line 263: Any hypothesis for why the lowest MDA8O3 occurs for PC3? Could it be related to the synoptic pattern of Fig. 4 and associated precipitation (not shown)? 23. Line 279: Delete "in the eastern region" 24. Fig. 8, Lin3 285: What constitutes "serious?" Perhaps it would be good

to plot the pollution threshold values here for O3 and PM2.5. Plotting these curves (they would be straight lines) would help the reader to identify how bad (or good) the air quality actually is in terms of PM2.5 and O3. The authors discuss "over-standard" rates, so a threshold must have been used (plot it). I believe these values are 160 and 75 ug/m3 for O3 and PM2.5, respectively... 25. Line 286: For (2), over-standard rates are not plotted – concentrations are. Please clarify. If the authors are suggesting that O3 an PM2.5 concentrations are similar between PCs, then please reword. 26. Line 287: For (3), it looks like Type 4 is leading, not Type 1, for O3 concentrations from 0900-1500. Since this is when concentrations are largest, the "Type 1 > Type 4 > ..." may mischaracterize your argument. 27. Line 302: "Activities"? Do the authors mean "modulations"? This is unclear.. 28. Line 308: "Makes summer always hot and moist" grammar needs revisions 29. Line 316: "presents negative" should be followed by "(Fig. 9a)" to guide the reader. Also, why are Tmax anomalies negative under this PC? 30. Lines 312-321: Precipitation is integral to the lifecycle of PM2.5 and O3. The linkages between the precipitation anomalies and Fig. 4 should be explicitly discussed. I believe the authors attempt to do this in Sec. 3.2, but that discussion is more appropriate here. 31. Lines 328-331: This sentence is unclear and should be revised. Also, there is an instance here where the authors use an acronym in one part of the sentence but not the other. Please be consistent. Also, how do negative FLWD anomalies result in a deeper PBL? 32. Sec. 4.1, P3: I believe that stability should be discussed here in addition to a more detailed discussion of precipitation "anomalies" associated with each PC. Thermal stratification of the PBL will dictate the mixing depth of the PBL and thus regulate the pooling of these aerosol/pollution plumes. Looking at the correlation between Tmax, PF, FLWD, etc. is not enough. 33. Lines 346-349: Here is a wonderful chance to discuss what is special about PC4 on a synoptic level. Why is PC4 leading to the largest O3 events synoptically? Do these same conditions favor the co-occurrence of O3 and PM2.5? 34. What is the difference between the Yangtze River and the YRD? These should be labeled on a map for readers... 35. It seems as though there is a window of moisture availability that is large enough to allow

hygroscopic growth of PM2.5 but is sufficiently small to allow for the important photo-chemical processes that regulate O3. It would seem to me that identifying this moisture window, as well as its sensitivity to PCs, would be a very significant contribution and I recommend that it be studied further to more precisely identify the PCs responsible for co-occurring O3/PM2.5 events. Identification of this moisture window can be based on an optimal relative humidity for compound pollution events too. This window can change by region and PC type. 36. Line 368: Strengthens compared to Type 1? Type 2's trough does not necessarily strengthen from the Type 1 pattern. Please reword. 37. Lines 357-387: The authors give the percentage of days with compound pollution for types 1 and 2. However, this does not elucidate which type is more efficient at producing compound pollution. The authors should include the percentages of compound pollution days for types 3 and 4. From the results here, I'd suspect that types 1 and 2 are the most efficient compound pollution setups, but this can be confirmed by including the percentages as for types 1 and 2. 38. Lines 396-397: These percentages need to be presented for Types 3 and 4 as higher percentages may indicate that PCs 3 and 4 are more efficient setups for co-occurring pollution events, even if the PCs occur less frequently. 39. Line 398: "line" should be "axis" 40. Line 400: Again, what is "Meiyu" season for non-local readers? 41. Lines 400-401: How do higher temperatures suppress O3 production? I would suspect that the higher relative humidifies are primarily responsible.... 42. Line 403: Is the low pressure trough at the surface or at 500 hPa? 43. Lines 402-404: Again, this "small amount of water vapor transport" suggests that there is a nominal vapor pressure deficit conducive to compound pollution events. In an environment of appropriate stability and PBL characteristics, compound pollution may be especially severe. 44. Lines 407-408: It appears that the WPSH only shifts north in your objective PC analysis, not southwards and eastwards...can the authors clarify? 45. Line 411: Why does water vapor lead to a sink of O3? Water vapor by itself cannot remove O3 from the atmosphere or prevent its formation. Are the authors referring to the supersaturation, dew point depression, etc.?

References

Bluestein (1992): Synoptic-Dynamic Meteorology in Midlatitudes: Principles of Kinematics and Dynamics, Vol. 1 1st Edition Rahimi et al. (2018): Exploring a Variable‐Resolution Approach for Simulating Regional Climate Over the Tibetan Plateau Using VR‐CESM. J. Geophys. Res.
* * *

---

## Author Comment (AC2) · 4 Jan 2021

Dear referee,

Thank you for your efforts for reviewing our manuscript. We highly appreciate to receive your useful comments. These comments are very constructive, and we have now further revised our manuscript in light of all the comments. Based on the helpful suggestions, we believe that we should have addressed questions and your concerns appropriately, and adequately. Please find our point-by-point responses in the supplement.

Please also note the supplement to this comment:

https://acp.copernicus.org/preprints/acp-2020-596/acp-2020-596-AC2-supplement.pdf

---

## Author Response (AR1)

Dear Editor,

Thank you for your efforts for handling our manuscript. We appreciate to receive the useful comments from all reviewers. These comments are very constructive, and we have now further revised our manuscript in light of all reviewers' comments. Based on the helpful suggestions from all reviewers, we believe that we should have addressed questions and concerns from all reviewers appropriately, and adequately. Please find our point-by-point responses below.

**Anonymous Referee #1:**

It is appreciated that the authors took great efforts to elucidate the complex relationship between WPSH and air pollution in China during summer, which is interesting. However, the manuscript was not well organized (synoptic characteristics and their influences on pollution are better given in a section), and most analyses were superficial without observational evidence. The manuscript needs better organization and careful English editing.

RESPONSE: Thank you for your valuable time to review this manuscript. We totally agree with your comments. Therefore, we have reorganized our writing structure thoughtfully in the current version. Particularly, we have carefully provided more information regard to the descriptions associated with synoptic weather characteristics and their influences on pollution. We have also added more information in order to provide in-depth results with observational evidences as suggested.

1. Only four types were identified, which tends to oversimplify the complex synoptic situations during summer in eastern China. By contrast, six types were identified by Han et al. (2020) to understand the influence of synoptic weather on the summertime O3 pollution in eastern China; nine types were classified by Ye et al. (2016) for the aerosol pollution in the North China Plain. The region selected for the classification should be consistent with the studied region (eastern China in Fig. 1). Why use the region shown by the black squares in Fig. 4? And present it in a larger domain in Fig.4? The classified region cannot resolve the processes needed? The selection of the classified region can significantly influence the classification results.

RESPONSE: Many thanks for your kind suggestions. The classification results can be largely influenced by the selected region, seasonal effects, and temporal length (time-series) of samples. We have read through the suggested references from reviewer. We found that Han et al. (2020) objectively identified six predominant synoptic weather patterns (SWPs) at 850 hPa level over eastern China (i.e., 20-40°N, 110-130°E) in the summer during 2013–2018. Ye et al. (2016) applied statistical analyses to identify nine types of synoptic flow based on sea level pressure over the North China Plain during autumn and winter (2004–2014). The settings are quite different from our study. Particularly, we chose a relatively large spatial extent as our classified region (0-60°N, 65-150°E) to capture the characteristics of the West Pacific Subtropical High (WPSH) at 500 hPa, as this can modulate the weather conditions and resultant co-occurring pollution events over eastern China. In this study, the number of SWP is determined by explained cluster variances (ECV) (Hoffmann & SchlüNzen, 2013; Philipp et al., 2014). The number of synoptic patterns (k) is optimized when the ΔECV is at the highest value, which suggests that the performance of classification has been improved substantially and with stability (Ning et al., 2019).

In order to further address the comments, we have now included a Fig. S1 to present the

changes of ΔECV against different numbers of classified synoptic patterns, including how we have optimized four SWPs. We have also marked the location of our classified region on the revised Fig. 4. Furthermore, we have revised our title as "Large-scale synoptic drivers of co-occurring summertime ozone and PM$_{2.5}$ pollution in eastern China".

Reference:

Han, H., Liu, J., Shu, L., Wang, T. and Yuan, H.: Local and synoptic meteorological influences on daily variability in summertime surface ozone in eastern China, Atmos. Chem. Phys., 20(1), 203–222, doi:10.5194/acp-20-203-2020, 2020.

Hoffmann, P. and Heinke SchlüNzen, K.: Weather pattern classification to represent the urban heat island in present and future climate, J. Appl. Meteorol. Climatol., 52(12), 2699–2714, doi:10.1175/JAMC-D-12-065.1, 2013.

Ning, G., Yim, S. H. L., Wang, S., Duan, B., Nie, C., Yang, X., Wang, J. and Shang, K.: Synergistic effects of synoptic weather patterns and topography on air quality: a case of the Sichuan Basin of China, Clim. Dyn., 53(11), 6729–6744, doi:10.1007/s00382-019-04954-3, 2019.

Philipp, A., Beck, C., Esteban, P., Krennert, T., Lochbihler, K., Spyros, P., Pianko-kluczynska, K., Post, P., Alvarez, R., Spekat, A. and Streicher, F.: Cost733 user guide., 2014.

Ye, X., Song, Y., Cai, X. and Zhang, H.: Study on the synoptic flow patterns and boundary layer process of the severe haze events over the North China Plain in January 2013, Atmos. Environ., 124(January 2013), 129–145, doi:10.1016/j.atmosenv.2015.06.011, 2016.

2. The classification results are odd. For example, in July of 2016, it quickly turned form Type 2 to Type 4, and then became Type 1 during a few days (Fig. 5). The WPSH cannot jump like that (e.g., from Type 2 to Type 4 in 24 hours). Besides, almost the whole June of 2016 was identified as Type 1 ("South BTH-North YRD O$_3$-PM$_{2.5}$ compound pollution"), while the South BTH and North YRD often experience clean and pollution situations during a few days (https://www.aqistudy.cn/historydata/daydata.php?city=%E5%8D%97%E4%BA%AC&month=201606). The synoptic pattern of Type 1 cannot explain the formation and evolution of pollution in South BTH and North YRD. The variations of pollution level may be primarily controlled by other synoptic weather systems at 900/850-hPa, rather than the 500-hPa WPSH. The same problems also existed in other Types/regions. The classification results and their relationships with pollution are suspicious and unreliable.

RESPONSE: Thank you for pointing out the "odd" results. In our study, the movement of the WPSH is generally affected by the weather phenomenon of its surrounding climatic systems (such as typhoons, the Tibetan high, etc.) (Ge et al., 2019; Liu and You, 2020; Shu et al., 2016). In order to address this comment, we have now included a Fig. R1 to show the synoptic weather patterns at 500hPa between July 05, 2016 and July 08, 2016. Our results showed that the weather pattern has first quickly changed from Type 2 to Type 4, then finally became a Type 1 pattern under the influence of Super Typhoon Nepartak (Fig. R2). Particularly, our statistical analyses showed that Type 1 pattern can represent the conditions of compound pollution events. Under the influence of this large-scale synoptic pattern, there is a relatively high probability of the occurrence of compound pollution in the BTH-NYRD region, with the maximum occurrence probability reaching to 55.09%. These variations of air pollution level may be primarily controlled by other synoptic weather systems at 900/850-hPa closely related to the movement of WPSH. Additionally, moisture driven by the WPSH could create

**inconducive condition for the O$_3$ formation across South China(Zhao and Wang, 2017). Furthermore, the amount of water vapor is largely reduced as it has been transported to North China through airflow trajectories, resulting in a moderation of RH and high temperature over North China which enhance the formation of O$_3$-PM$_{2.5}$ co-occurring event.**

**More importantly, one aim of our study was to examine the effect on O$_3$-PM$_{2.5}$ co-occurring event caused by the large-scale WPSH related synoptic pattern. As WPSH at 500-hPa is easily identified based on 5880 gpm contour line, we applied the following condition (500 hPa GH) as our dependent variable for classification. In order to further address this comment, our revised Fig.R3 has shown the synoptic weather pattern at 850 hPa corresponding to the classification. This figure can represent matching characteristics of the WPSH at 850 hPa and 500 hPa (Fig.4 and Fig. R3). Based on the comparison, using 500hPa GH for classification should be accurate and reliable in this study.**

Reference

Ge, J., You, Q. and Zhang, Y.: Effect of Tibetan Plateau heating on summer extreme precipitation in eastern China, Atmos. Res., 218, 364–371, doi:10.1016/j.atmosres.2018.12.018, 2019.

Hoffmann, P. and Heinke SchlüNzen, K.: Weather pattern classification to represent the urban heat island in present and future climate, J. Appl. Meteorol. Climatol., 52(12), 2699–2714, doi:10.1175/JAMC-D-12-065.1, 2013.

Huth, R.: An intercomparison of computer-assisted circulation classification methods, Int. J. Climatol., 16(8), 893–922, doi:10.1002/(SICI)1097-0088(199608)16:8<893::AID-JOC51>3.0.CO;2-Q, 1996.

Liu, J. and You, Q.: A diagnosis of the interannual variation of the summer hydrometeor based on ERA-interim over Eastern China, Atmos. Res., 231(October 2018), 104654, doi:10.1016/j.atmosres.2019.104654, 2020.

Philipp, A., Beck, C., Esteban, P., Krennert, T., Lochbihler, K., Spyros, P., Pianko-kluczynska, K., Post, P., Alvarez, R., Spekat, A. and Streicher, F.: Cost733 user guide., 2014.

Shu, L., Xie, M., Wang, T., Gao, D., Chen, P., Han, Y., Li, S., Zhuang, B. and Li, M.: Integrated studies of a regional ozone pollution synthetically affected by subtropical high and typhoon system in the Yangtze River Delta region, China, Atmos. Chem. Phys., 16(24), 15801–15819, doi:10.5194/acp-16-15801-2016, 2016.

[Figure]

**Fig. R1. the case during July 5-8,2016**

[Figure]

**Fig. R2 Track of Super Typhoon Nepartak. This figure can be downloaded on the website (https://sharaku.eorc.jaxa.jp/TYP_DB/data/TYP_DB_COMMON/track/bst_2016s.02W.NEP ARTAK.jpg).**

[Figure]

**Fig. R3. The 10-m wind (vectors; see scale arrow at the bottom right in units of 5 m/s) and 850-hPa GH (contours; see scale bar at bottom in units of gpm) patterns based on objective classification**

3. Physically, the synoptic weather patterns influence the pollutions via PBL structure (e.g., large-scale subsidence), regional-scale transport of pollutants (e.g. PM2.5 and VOCs), and occurrence of precipitations. All these physical processes underlying were not well analyzed. The authors cannot just use simple correlation analysis to explain the mechanisms, which makes the conclusion unreliable and inconvincible. At least, typical pollution episodes of each synoptic weather type should be analyzed in-deep with observational evidence to validate the hypothesis given.

**RESPONSE: Based on your suggestions, this study has now considered the role of boundary layer structure in modulating the pollution. Particularly, we conducted an in-depth analysis of the compound pollution under Type 1 and Type 2 patterns. The underlying physical mechanism would be explained by our investigation regarding the characteristics of boundary layer structure, precipitation and ground-level wind flow over the polluted regions. More descriptions have now been noted on lines 410-424. Please also find the information below.**

**"Furthermore, we summarized boundary layer structure, precipitation, and ground-level wind flow across the BTH region. Based on the characteristics, we separately defined Type 1 and Type 2 into clean (both concentrations of the $O_3$ and $PM_{2.5}$ are less than polluted level) and compound pollution periods (Figs. 11 and S10-S11). Particularly, Type1 has a significantly warmer temperature over the boundary layer during the compound pollution period of BTH region than that of the clean period. The daytime BLH under compound pollution condition was also higher than that of the clean condition. In addition, there were different directions of**

prevailing during the two periods, which prevailing winds during the compound pollution period were usually southward and could be driven by air pollutants transported from the southern plains (Fig. 11; see also Miao et al., 2019b, 2020). Co-influencing by the topographical effect of the northern mountainous areas, air pollutants could be trapped in the BTH region. In comparison, although there was southward wind prevailing in the BTH region (Figs. 11 and S11), the rain belt also located in the southern area of BTH might lead to the potential removal of PM$_{2.5}$ (Fig. 9j). Therefore, compound pollution across the BTH region might mainly be due to local emissions of air pollutants."

[Figure]

Fig. 11. The daily variation of horizonal wind, potential temperature (PT) and BLH of boundary layer in the BTH under clean and compound pollution period of Type 1 and Type 2 (a, b, e, f). The vertical cross-section of u-wind, w-wind and PT for the same situation of BTH (c, d, g, h). The w-wind is multiplied by 100 when used. The data has been derived from ERA5 reanalysis data.

4."The BLH was calculated according to the method given by Guo et al. (2016, 2019), and the FLWD [frequency of light wind (< 2 m s−1) days], precipitation frequency (PF), and MDA8 O3 were also counted." The detailed information about BLH should be directly given. At present, only a few cities have afternoon soundings during summer. Only 08:00 and 20:00 LT soundings were used to calculate the BLH, which is inappropriate for this study. How to calculate the FLWD and PF? Why only choose these specific parameters? Are they significantly correlated to the pollution levels in all the studied regions (e.g., BTH, YRD, GZP)? How about the precipitation intensity and amount, and its lasting time? How about the wind directions and wind shear, and associated transport of pollutants (PM2.5 and VOCs)?

RESPONSE: Thanks for the comments. More information regarding BLH has now been added in supplementary materials. Please find the details as follows:

"The bulk Richardson number (*Ri*) was applied to calculate the BLH.

Ri can be expressed as:

$$R(i) = \frac{(g/\theta_{vs})(\theta_{vz}-\theta_{vs})(z-z_s)}{(u_z-u_s)^2+(v_z-u_s)^2+(bu_*^2)} \tag{1}$$

where $z$ is height above ground, $s$ is the surface, $g$ is the acceleration due to gravity, $\theta_v$ is virtual potential temperature, $u$ and $v$ are the component of wind speed, and $u_*$ is the surface friction velocity. Due to a much smaller magnitude in compared with bulk wind shear term in the denominator, this study does not need to consider the influence of $u_*$ (Seidel et al., 2012).".

Furthermore, our revised study has now applied 14:00 LT soundings to estimate the BLH, while the results of BLH at 08:00 and 20:00 LT soundings (calculated in original version) were reported in supplementary materials as references (Fig. S13). Specifically, we have added the following variables for calculation: FLWD and PF. Specifically, FLWD is the frequency of light wind ($< 2$ m s$^{-1}$) days, which can be defined as the ratio between the number of the days with average daily WS lower than 2 m s$^{-1}$ and the total days of each synoptic pattern; PF is precipitation frequency, which can be defined as the ratio of the number of the rainy days to the total days associated with each synoptic pattern). The above definitions were due to an unfavorable condition for transporting air pollutants during low WS days as well as low possibility of forming $O_3$-$PM_{2.5}$ co-occurring pollution during low PH conditions. We also applied specific meteorological parameters associated with air pollution levels in this study, such as Tmax, RH and BLH. Based on our results, we also found that Tmax can exceed 27°C during high $O_3$ events across five urban clusters of the study area, and moderate RH can be associated with $O_3$-$PM_{2.5}$ compound pollution in the BTH region and North YRD area. Furthermore, high BLH can enhance the environmental capacity of atmospheric pollutants resulting in a reduction of air pollutant concentration. Additionally, there was not a significant difference of the average daily precipitation among the four types of synoptic weather patterns (Fig. R4). However, we have found a significant difference of precipitation level during clean and compound pollution period under the same SWP (see Figure S10 and S11). Due to a lack of hourly precipitation data, this study cannot show the intensity and lasting time. Therefore, we have now added more information regarding this in the limitation section, as well as a suggestion regarding the future improvement with high-temporal-resolution data.

[Figure]

Fig. R4. The daily mean amount of precipitation under each SWPs.

[Figure]

**Fig. S10. Precipitation, WS, and WD during clean and compound pollution period under Type 1 over BTH.**

[Figure]

**Fig. S11. Same as Fig. S8 but for Type 2.**

5. In the Results and Discussion sections, the "rain belt", "Meiyu", "rain band", "heavy precipitation" were simply analyzed with very few observational evidence. Since this study focuses on the movement/evolution of WPSH in summer (rainy season), more in-deep analysis on the links between precipitation and pollutions should be given with observational evidence and typical episodes.

**RESPONSE: Thanks for your suggestion. It is indeed our limitation as the position of the rainband could only be sketchily plotted based on the frequency and volume of precipitation under each SWP (Fig. 8i-l). Therefore, this study aimed to characterize the trend and location of rain belt in relation to the movement of WPSH in summer, based on separate discussions with each SWP. More information can be referred as follow: "Type 1 is characterized by humid condition in the southern area and dry condition in the northern region owing to an extensive southwestern flow of the WPSH, resulting in a rain belt found in southeastern coastal area such as PRD and YRD regions. Type 2 is associated with meridional flow and dry and wet anomalies in northern China, resulting in a rain band locating at the central areas of between BTH and YRD regions due to the northern advance of the WPSH compared with Type 1. Furthermore, there is a greater RH for most of the study sites under Type 3 and Type 4, possibly a result of the shifted rain belt in the BTH and NEM regions under Type 3 once the northern boundary of the WPSH reaching at 37.5°N, and an occurrence of heavy precipitation across the western PRD region as well as central areas of between BTH and YRD regions under Type 4 (Fig. S6).".**

**Additionally, we have now shown the linkages between precipitation and compound pollution in Figs. S10 and S11 as well as the response of the last question, with evidences showing that precipitation amount during clean period is greater than that of the polluted period over the BTH region.**

[Figure]

**Fig. 8. Same as Fig. 6 but for Tmax (a–d), RH (e–h), and PF (i–l). The black solid line presents the rain belt of each SWP.**

6.Section 4.2 "Effects of NO2 on O3". The best proxy of photochemical reactivity is the ozone potential efficiency (OPE) but not the ratio of O3 to NO2. High photochemical reactivity probably appeared with high O3 and NO2 concentration but reasonably with a low O3/NO2 ratio. It is not accurate to take the O3/NO2 ratio as the judging criteria.

**RESPONSE: Many thanks for your kind suggestion. Indeed, ozone potential efficiency (OPE)**

is the best proxy of photochemical reactivity, however, the related data for calculation was unavailable. As an alternative, previous studies have demonstrated that the photochemical reaction of NO, NO2, and $O_3$ in the troposphere form could a closed system(Yu et al., 2020), and this photochemical cycle of NOx and $O_3$ could be the basis of photochemical processes in the troposphere. Therefore, oxidant (Ox, Ox=$O_3$+$NO_2$), a conservative quantity over a short temporal scale, could be an alternative parameter to evaluate photochemical processes because NO can quickly react with the equivalent amount of $O_3$ to generate $NO_2$(Kley et al., 1994). Particularly, some studies have demonstrated that Ox could be used to represent OPE (Chang et al., 2020; Ge et al., 2013). Therefore, we used Ox to be an alternative of OPE, not the ratio of $O_3$ to $NO_2$ in the last manuscript.

Based on these assumptions, we further revised our content. For example, revised, "Figure 10 can present the daily variation of $NO_2$ and Ox. These include daily variations of $NO_2$ showing two peaks during a day, including a first peak at the morning and an second peak associated with traffic emissions in the evening (Xie et al., 2016; Yu et al., 2020). As we found the lowest point of $NO_2$ at 15:00 (BJT), and $NO_2$ can be photolyzed to produce $O_3$ during the day, this study assumed that this particular time was the peak formation ozone across the study areas. As $NO_2$ was consumed through a photochemical reaction with the involvement of other precursors to produce a large amount of $O_3$, Ox could form a peak during the afternoon. In particular, abundant sunlight in summer is beneficial to the photochemical reaction process, but since most parts of eastern China are in a subtropical climate with the same period of rain and heat, the existence of the rainy season will inevitably inhibit the summer photochemical process. Under different SWP, the photochemical reaction over each area has an obvious relationship with the rain belt. For example, the rainy season in BTH and NEM areas mainly occurs in Type 3, and the Ox of Type 3 in this area is significantly lower than other SWPs.". Therefore, we have changed the section title from "Effects of $NO_2$ on $O_3$" to "Potential implications of $NO_2$".

[Figure]

Fig. 10. Daily variation of $NO_2$ and Ox under four SWPs in key urban clusters.

[Figure]

**Fig. 1. Average concentration of MDA8 O₃ (a) and PM₂.₅ (b) in eastern China during summers of 2015–2018. Stations and key urban clusters (black boxes) are shown in (a).**

3. Line 165-166, "More detailed information about the T-PCA method can be found in Miao et al. (2017)." The detailed information of the method should be directly given.

**RESPONSE: More detailed information regarding the T-PCA method is now added to lines 191-197 on pages 7-8:**

**"The cost733class is a FORTRAN software package consists of several modules for classification, evaluation and comparison of weather and circulation pattern. First, T-PCA classification of the cost733class performs spatial standardization on weather data. Then split data to 10 subsets and estimates principal components (PCs) of weather information based on singular value decomposition, the PC score for each subset can be calculated after oblique rotation, and compares 10 subsets based on contingency tables to select the subset with highest sum and return its types (Miao et al. 2017)."**

4. Some literatures were not properly cited. Please carefully check the citations of the whole manuscript. Some are given below. "In general, PM2.5 pollution is featured with obvious diurnal and seasonal changes. Due to the influence of atmospheric diffusion conditions such as precipitation and wind speed (WS), it tends to be enhanced in the morning and evening, lower at noon, and higher in winter and lower in summer." Dose the author mean that the diurnal variations of precipitation and wind speed modulate the pollution level? It is odd. How about the emission and PBL?

**RESPONSE: Thanks for your comments. We have revised our content as follow: "In general, a significant diurnal variation of PM₂.₅ pollution was observed, possibly due to obvious the local emissions caused by industrial production and human activities for daily living (Amil et al., 2016; Liu et al., 2019). Particularly, the pollution level was higher during the morning and evening of a normal weekday, with a weakening effect found in the afternoon which may be caused by the co-effects of boundary layer structure as well as anthropogenic emissions. There was also a seasonal variation of PM₂.₅ pollution across China, indicating a higher level of pollution in winter than summer (Ye et al., 2018; Zhang and Cao, 2015)". (lines 66-72 of page 3).**

5. "Summer O3 pollution has gradually been prominent, replacing PM2.5 as the primary pollutant in the air…" Is it true? At present, PM2.5 still is the dominant pollutant in China.

**RESPONSE: We have revised the sentence as follow: "While PM₂.₅ is still one of the dominant air pollutants across China, surface O₃ pollution in summer has gradually been prominent. Several studies even indicated that O₃ might have replace the role of PM₂.₅ as the primary air pollutant during summer (Li et al., 2019).".**

6. "Miao et al. (2015) suggested that strong northwesterly synoptic winds, low BLH (boundary layer height), high RH and stable atmosphere are more prone to aerosol pollution in the BTH region during wintertime …" The strong northwesterly winds would favor the dispersion of pollutants in winter.

**RESPONSE: Thank you for the information. We have revised the content as follow: "Miao et al. (2015) suggested that low boundary layer height (BLH) and stable atmosphere would be an unfavorable condition for the dispersion of winter aerosol pollution over the BTH region.".**

7. "Shi et al. (2020) studied the spatial distribution of O3-8h (O3 8-hour moving average) and PM2.5, and their sensitivity of meteorological parameters; pronounced positive (negative)

correlation between temperature (BLH and absolute humidity) and O3-8h was found, but the relation between WS and O3-8h was spatially different; for PM2.5, it was negatively (positively) correlated with temperature, WS and BLH (absolute humidity)." It is another inappropriate citation. Please carefully read the previous study (Shi et al., 2020) and properly introduce it.

**RESPONSE: Thank you for the information. We have revised the content as follow:**

**" Shi et al. (2020) studied the sensitivity of O$_3$-8h (O$_3$ 8-hour moving average) and PM$_{2.5}$ associated with meteorological parameters. This study focused on the air pollution and meteorological conditions between January and July, 2013, with a result showing that temperature could have the greatest impact on the daily maximum O$_3$-8h, while the PM$_{2.5}$ sensitivities are negatively (positively) correlated with temperature, WS, and BLH (absolute humidity) in most regions of China.".**

8. "Recently, Han et al. (2020) revealed that meteorological factors can explain ~46% of the daily variability in summertime surface O3, while synoptic factors contribute to ~37% of the overall meteorological effects on the daily variability of surface O3 in eastern China." More detailed information on Han et al. (2020) should be presented since it is quite similar to this study, such as its studied period, method, and classification results?

**RESPONSE: Thanks for your comments. More details regard to the results of Han et al. (2020) have now been presented in the revised manuscript (lines 110-116 of page 5):**

**"Recently, Han et al. (2020) assessed the impacts of local and synoptic meteorological factors on the daily variability of surface O$_3$ over eastern China. This study revealed that the meteorological factors could explain ~46% of the daily variations of summer surface O$_3$. Particularly, synoptic factors contributed to ~37% of the overall effects associated with the meteorological factors. Furthermore, six predominant SWPs were identified by the self-organized map, and the results indicated a weak cyclone system and a southward prevailing wind inducing a positive O$_3$ anomalies over the eastern China.".**

9. Line 95-97, "The abovementioned indicates that the variation of meteorological factors, which are mainly driven by the evolution of different weather circulation situations, play a non-negligible role in air pollution. Therefore, classification of air pollution according to the meteorological circulation has become particularly important …" The abovementioned literatures cannot support this statement.

**RESPONSE: Thank you for the comment. This statement has now been deleted in the revised manuscript.**

10. Line 101-102, "In recent years, it has become possible to objectively classify atmospheric circulation conditions using weather data such as GH, sea level pressure…" It is not true. The objective classification method has been used since the 1990s.

**RESPONSE: Thank you for pointing out the biased description. We have now revised it based on your suggestion (Lines 121-123 of page 5).**

11. Line 104-105, "the objective approach has been widely used in air pollution research (Beck & Philipp, 2010)…" Beck and Philipp (2010) didn't study the pollution issues.

**RESPONSE: We apologize to the inappropriate citation. This citation is now removed from our manuscript.**

12. Line 115-118, "Many studies have suggested that PM2.5 and O3 pollution are mainly related to the East Asian summer monsoon (EASM) and western Pacific subtropical high (WPSH) (Li et al., 2018a; Xie et al., 2017; Yin et al., 2019; Zhao et al., 2010)." More detailed information about these previous studies can be given, their studied periods, temporal scale and spatial scale. Seasonal variation? Inter-annual variation? The paper of Xie et al. (2017) has been withdrawn, please check (https://acp.copernicus.org/preprints/acp-2017-500/).

**RESPONSE: Thank you for your kind suggestion. We have revised the content as follow:**
**"Many studies have suggested a moderating effect of East Asian summer monsoon (EASM) and western Pacific subtropical high (WPSH) on air quality over China (Li et al., 2018; Yin et al., 2019; Zhao et al., 2010). In particular, Li et al. (2018) applied RegCM4-CHEM simulation to analyze the differences of ozone level among three strong and weak monsoon years, and found that the concentrations of $O_3$ over the central and eastern China were higher in strong EASM years than that in weak EASM years.".**

13. Line 129-131, "the compound O3-PM2.5 pollution-related meteorological conditions, should be complex and likely to be associated with certain weather types". This statement is not well supported.

**RESPONSE: Thank you for your suggestion. We have revised the content as follow: "Due to a variability of local meteorological conditions under the impacts of various synoptic weather types and modulation of large-scale WPSH movement, the causes and consequences of meteorological factors for the formation of compound $O_3$-$PM_{2.5}$ pollution could be complex.".**

**Anonymous Referee #2:**

This paper investigated the co-occurring of ozone and PM2.5 pollution of eastern China in summer. Four synoptic weather patterns (SWPs) were detected and the air pollution feature under each SWP was analyzed. The paper is not well organized and difficult to follow. The authors should revise the manuscript carefully to meet the standard of ACP. The detailed comments are listed below.

**RESPONSE: Thank you so much for your valuable comments. We have carefully addressed your concerns. We have also reorganized our manuscript thoughtfully as suggested.**

Major comments:

This paper investigated the co-occurring of ozone and PM2.5 pollution in summer. However, the probability of PM2.5, ozone and compound pollution in each site could not be found in the manuscript. Please show their distribution at beginning of the manuscript. Their distribution under each SWP should be also shown to see the impacts of different weather patterns.

**RESPONSE: Thanks for your suggestion. We have now included a figure S11 (supplementary material) to represent the calculation for the probability of compound pollution. Particularly, we found significant $O_3$-$PM_{2.5}$ compound pollution events over five urban clusters during the study period, especially in the BTH region. Furthermore, BTH-NYRD under Type 1 and BTH region under Type 2 could be concluded as a result of $O_3$-$PM_{2.5}$ co-occurring pollution due to a greater occurrence of compound pollution. More information regarding spatial distributions of $O_3$, $PM_{2.5}$ and compound pollution under each SWP were given in Fig. 12.**

[Figure]

**Fig. S12.** The number (a) and probability (b) of occurrence of compound pollution days under all SWPs in each site, (c) and (d) is the same as (b), but for Type 1 and Type 2.

[Figure]

**Fig. 12. Schematic diagrams describing the relationships between the WPSH, four SWPs and summertime O₃ and PM₂.₅ pollution in various regions.**

In this work, four synoptic weather patterns were detected and their difference were compared. The physical understanding the SWPs could be improved. In my opinion, Type 1 and type 2 represents normal WPSH pattern during early and late summer, respectively. Type 3 and type4 reflects two splitting states (southern mode and northern mode) of the WPSH, which mainly occurs in late summer. The climate background of early and late summer is quite different from each other. Thus, the SWPs of early and late summer should be compared separately.

**RESPONSE: Thank you for your comments. For most of the days, Type 1 and Type 2 can represent both regular WPSH pattern during early and late summer. Type 3 and type 4 may demonstrate the two splitting states (southern mode and northern mode) of the WPSH, which mainly occurs in the late summer. Particularly, characteristics of circulation of these synoptic patterns have been described and compared in section 3.2:**

**"The location of Type 1 related western ridge point and northern boundary of the WPSH at 500 hPa is around 120°E and 30°N, respectively, (Fig. 4a and Table S2). The southwestern flow of this WPSH could chemically transport water vapor to the YRD region, resulting in a southwestward prevailing wind across the YRD region and westward flow from the north of the WPSH forming a convergence area at 850 hPa. These conditions were also associated with high temperature and humidity during the summer with Meiyu season, which Meiyu season is a climate phenomenon with continuous cloudy and rainy days generally occurring during June and July every year in the middle and lower reaches nearby Yangtze river, Taiwan of China, central/southern Japan, and southern Korea. For Type 2, the westerly trough could deepen as the WPSH shifts northward slightly from Type1 or retreats southeast from Type 3 (Fig. 4b). The southerly wind from the ocean could interact with the northern periphery of the WPSH. As a result, the sea-land interaction could affect the southeastern region across China,**

while northern China could be mainly controlled by the westerly trough. In compared with Type 2, the impacts of Type 3 could shift the boundary of WPSH to a higher latitude, with a westward extension (Fig. 4c), which disintegrating the closed high-pressure monomer along the eastern coast of China and remaining the main body of the WPSH over the ocean (Figs. 4c and S4). This has led to a condition completely controlled by the monomer of the WPSH over the YRD region, resulting a hot and dry weather at the end the rainy season at the beginning of mid-summer. Figure. 4d indicated that the location of the WPSH monomer was more northern and western with respect to other SWPs, controlling northern China for a long time, the western ridge point was around 95°E and the northern boundary is around 40°N.

Figure 5 presents the daily and annual variations of the SWPs in the summers of 2015–2018. The advance of the WPSH in eastern China occurs in June and July, while a gradual withdrawal of the WPSH mainly occurs in August. Particularly, Type 1 and Type 2 represent normal WPSH characteristics during early and late summer. Type 3 and Type 4 could reflect a split of the WPSH, which mainly occurs in late summer. Consequently, there were 167, 117, 52 and 32 days for the Type 1, Type 2, Type 3 and Type 4 over the study period, respectively. Since WPSH movement is generally affected by the weather phenomenon of its surrounding climate systems (such as typhoons, the Tibetan high, etc.) (Ge et al., 2019; Liu and You, 2020; Shu et al., 2016; Wang et al., 2019), it could result in a short-term southward retreat during the advancement of WPSH (e.g., around 10 August 2018) and a short-term northward advance during its process of retreat (e.g., 21 and 29 August 2016). In general, the WPSH could represent the evidences of intra-seasonal and interannual changes over China, which will inevitably modify the changes of weather, climatic and environmental conditions in eastern China.".

More information for local meteorological factors under the influences of the above climatic background, including related environmental impacts, have now been noted in section 3.4.

The abstract is not well written and following information should be included: 1) The distinct feature of each synoptic weather pattern; 2) the pollution features under each SWP; 3) which SWP mostly favors the co-occurring of ozone and PM2.5 pollution; 4) where the co-occurring of ozone and PM2.5 pollution happens.

RESPONSE: Thanks for your suggestion. We have revised the abstract to be more focusing on the co-occurrence of $O_3$ and $PM_{2.5}$ pollution under Type 1 and Type 2 synoptic weather patterns, including more details in the results. The revised abstract is as follow: "Surface ozone ($O_3$) pollution during summer (June-August) over eastern China has become more severe, resulting in a co-occurrence of surface $O_3$ and $PM_{2.5}$ (particulate matter with aerodynamic diameter ≤ 2.5 μm in the air) pollution recently. However, the mechanisms regarding how synoptic circulation pattern could influence this compound pollution remains unclear. This study here applied a T-mode principal component analysis (T-PCA) method to objectively classify the occurrence of four synoptic weather patterns (SWPs) over eastern China, based on geopotential heights at 500 hPa during summer (2015-2018). Four SWPs of eastern China are closely related to the western Pacific subtropical high (WPSH), exhibiting, significant intraseasonal and interannual variations. Based on ground-level air quality information and meteorological observations, remarkable spatial and temporal disparities of surface $O_3$ and $PM_{2.5}$ pollution were also found under the impacts of the four SWPs.

**Particularly, there were two SWPs sensitive to compound pollution (Type 1 and Type 2). Type 1 is characterized by a stable WPSH ridge with axis at about 22°N and the rain belt located in the south of Yangtze River Delta (YRD). High temperature, moderate humidity and low precipitation occurred in the region from BTH to northern YRD (BTH – NYRD), resulting in a co-occurrence of $O_3$ and $PM_{2.5}$ pollution. Additionally, air pollutants can be transported by the prevailing southerly winds from southern plains and accumulated in the southern BTH, resulting in a worsen pollution. Type 2 exhibits a WPSH dominance (the ridge axis ~25°N) and rain belt (over the YRD) in a higher latitude compared with Type 1. High temperature, medium-high humidity and low precipitation over the BTH were the conducive factors related to the occurrence of the compound pollution events under Type 2. Furthermore, low boundary layer height (BLH) and high frequency of light-wind days (FLWD) could create favorable conditions for pollution maintenance. Overall, synoptic weather patterns have played an important role as driving factors of surface $O_3$-$PM_{2.5}$ compound pollution in a regional context. In addition to the impacts of local emissions, our results may provide further insights regarding how regional environmental changes due to co-occurrence of high $PM_{2.5}$ and high $O_3$ level may be driven by the effects of meteorological factors. Overall, our findings demonstrate the important role played by synoptic weather patterns in driving regional surface $O_3$-$PM_{2.5}$ compound pollution, in addition to the large quantities of emissions, and may also provide insights into the regional co-occurring high $PM_{2.5}$ and high $O_3$ level via the effects of certain meteorological factors.".**

Other comments:

1. Line 26: Please describe details of SWPs here.

**RESPONSE: The details of SWPs are given as follows: "Type 1 is characterized by a stable WPSH ridge with axis at about 22°N and the rain belt located in the south of Yangtze River Delta (YRD)." And "Type 2 exhibits a WPSH dominance (the ridge axis ~25°N) and rain belt (over the YRD) in a higher latitude compared with Type 1". See lines 30-37 of page 2 in the revised version.**

2. Line 30: How about the ozone pollution over the regions where are not controlled by the WPSH or the prevailing westerlies. Where are these regions?

**RESPONSE: Most of the regions we focus on are controlled by the WPSH or the prevailing westerlies under four SWPs except for PRD region under Type 3 and Type 4. In the revision, we have focused on the SWPs which are more prone to the compound pollution.**

3. Line 34: Please explain the meaning of "some local areas".

**RESPONSE: "Some local areas" refers to "the BTH region under impacts of Type 1 and Type 2". Thanks and revised.**

4. Line 35: Please clarify where the co-occurring surface O3 and PM2.5 pollution happens.

**RESPONSE: The co-occurring surface $O_3$ and $PM_{2.5}$ pollution happened in the BTH – NYRD regions under Type 1 and BTH region under Type 2.**

5. Line 35-36: How the WPSH affects the boundary layer height and frequency of lightwind days?

**RESPONSE: The area controlled by the WPSH usually have shallow BLH and high FLWD. In the revised manuscript, we have added the structural characteristics of the boundary layer to better explain the causes of compound pollution.**

6. Line 36: What does the "different roles" mean? Please provide some explanations.

**RESPONSE: It should be "an important role". Sorry for incorrect description and thanks.**

7. Line 118-120: In abstract, it is mentioned that high temperature, moderate humidity and slight precipitation favors the ozone formation. It is not consistent with the statements here.

**RESPONSE: Thanks for your comment. Indeed, hot and dry air can enhance the photochemical reactions of $O_3$. Eastern China has a monsoon climate with dry winters and humid summers, so moderate humidity actually presents negative anomaly for summers, and it could be considered as a dry summer. In addition, we also analyzed the RH condition under different pollution scenarios (Fig. S14), and found RH is low-medium when $O_3$-only pollution occurs, and moderate RH would favor the co-occurrence of $O_3$ and $PM_{2.5}$. Therefore, the results were not inconsistent. In order to address the comments, we have now revised this description as follow: "High temperature, moderate humidity and low precipitation occurred in the region from BTH to northern YRD (BTH – NYRD), resulting in co-occurring $O_3$ and $PM_{2.5}$ pollution".**

[Figure]

**Fig. S14. Box-Whiskers for the RH under compound pollution, clean, O3-only, PM2.5-only period. In the Box-Whiskers plot, the central box represents the values from the lower to upper quartile (25th to 75th percentile). The vertical line extends from the maximum to the minimum value. The middle solid line represents the median, and the red plus represents the outlier.**

8. Line 124: It is not consistent with the conclusions in abstract. In abstract, it is found that the warm moist flow brought by the WPSH result in co-occurring surface O3 and PM2.5 pollution.

**RESPONSE:**

**Sorry for the unclear description. We have revised our conclusion as follow: "On one hand, the appropriate warm moist flow brought by the WPSH can promote hygroscopic growth of the fine particulate matter in some local areas (i.e., BTH-NYRD under Type 1 and BTH under Type 2), resulting in the increase of $PM_{2.5}$ concentrations. On the other hand, transboundary $O_3$ was transported to these local areas at the same time, which may contribute to the co-occurring surface $O_3$ and $PM_{2.5}$ pollution"**

9. Line 148: I cannot find the position of the urban agglomeration in Figure 1a. Please check.

**RESPONSE: We have changed the Figure 1a, and the black boxes shown in Figure 1a indicate the locations of key urban clusters.**

[Figure]

**Fig. 1. Average concentration of MDA8 O$_3$ (a) and PM$_{2.5}$ (b) in eastern China during summers of 2015–2018. Stations and key urban clusters (black boxes) are shown in (a).**

10. Line 207: Please show the temporal variations of co-occurring events.

**RESPONSE: We have added the temporal variations of co-occurring events in Fig. 3. The asterisks indicate the co-occurred events under Chinese standard (WHO interim target 1, IT-1), and the circles indicate the co-occurred events under WHO IT-2 in Fig. 3. The specific description is "China has implemented strict policies for emission control, and the effects of these policies were remarkable. However, despite a decrease in PM$_{2.5}$ in the last five years, there was also an increase in ozone pollution over China (Fan et al., 2020; Sun et al., 2016), although "double-high" pollution reported on the weather scale has been decreased. As the limit of PM$_{2.5}$ concentration for pollution control is relatively loose in China, previous studies usually referred the interim target 1 (IT-1) of the World Health Organization (WHO) as the standard threshold. Our study pushed forward to the next stage, in which we used IT-2 of WHO (24-h average concentration of PM$_{2.5}$ is 35 μg m$^{-3}$) as our target limit to count the number of compound pollution days across each region. Based on this target, the pollution days for 4 SWPs were 194, 52, 16, 47, and 20, respectively (Fig. 3). These results indicated a severe situation of compound pollution that is still deserved a public attention."**

[Figure]

**Fig.3. Time series of PM$_{2.5}$ pollution levels in key urban clusters, the black dots indicate the co-occurred events. The asterisks indicate the co-occurred events under Chinese standard (WHO interim target 1, IT-1), and the circles indicate the co-occurred events under WHO IT-2.**

11. Line 229: Please focus on the position and strength of WPSH.

**RESPONSE: The location of WPSH is shown on Table S2. Description regarding WPHS is now added to lines 266-285 of pages 10-11: "The location of western ridge point and northern boundary of the WPSH at 500 hPa in Type 1 is around 120°E and 30°N, respectively (Fig. 4a and Table S2). …… For Type 2, the westerly trough could deepen as the WPSH shifts northward slightly from Type 1 or retreats southeast from Type 3 (Fig. 4b). …… In compared with Type 2, Type 3 presents the boundary of WPSH in a higher latitude with a westward extension (Fig. 4c), disintegrating a closed high-pressure monomer along the eastern coast of China and the main body of the WPSH over the ocean (Figs. 4c and S4). …… Figure. 4d indicated that the location of WPSH monomer was more western and northern with respect with other SWPs, controlling the northern China for a long time; the western ridge point was around 95°E and the northern boundary was around 40°N.".**

**Table S2. The location index of the WPSH under four SWPs.**

| WPSH | Type 1 | Type 2 | Type 3 | Type 4 |
|---|---|---|---|---|
| The western ridge point | 120°E | 127.5°E | 110°E | 95°E |
| The northern boundary | 30°N | 32.5°N | 37.5°N | 40°N |
| The ridge axis | 22°N | 25°N | 32.5°N | 37.5°N |

12. Line 229: It should be the north advance of WPSH.

**RESPONSE: Yes, WPSH is slight shifting northward. We have now revised the content (lines 274-275 page 10).**

13. Line 234: Type 1 and type 2 represents normal WPSH pattern during early and late summer,

respectively. Type 3 and type4 reflects splitting of the WPSH, which mainly occurs in late summer.

**RESPONSE: Thanks for your advice. Based on your suggestions, we have now added the following information to the revised manuscript: (lines 286-290): "The advance of the WPSH in eastern China occurs in June and July, while gradual withdrawal of the WPSH occurs mainly in August, Type 1 and Type 2 represent normal WPSH characteristics during early and late summer. Type 3 and Type 4 could reflect a split of the WPSH, which mainly occurs in late summer.".**

14. Line 260: Type1 mainly occurs in June and it explains why the O3 concentration is higher.

**RESPONSE: Yes.**

15. Line 295: Please show the spatial distribution of the co-occurring surface O3 and PM2.5 pollution under each SWP.

**RESPONSE: The spatial distributions of the co-occurring surface $O_3$ and $PM_{2.5}$ pollution under impacts of each SWP are shown in Fig.12.**

[Figure]

**Fig. 12. Schematic diagrams describing the relationships between the WPSH, four SWPs and summertime $O_3$ and $PM_{2.5}$ pollution in various regions.**

16. Line 300: Potential meteorological factors should be included in the results of this paper.

**RESPONSE: Thanks for your suggestion. We have now moved the corresponding information to the "Result" section. Please refer to lines 347-376 on pages 13-14.**

17. Line 305-310: This part should be mentioned in introduction.

**RESPONSE: Thanks for the comment. We have moved this part to the introduction. Please refer to lines 93-110 on pages 4-5.**

18. Line 315-316: It is because the type 1 mainly occurs in early summer.

**RESPONSE: We have added more information to describe the occurrence of type 1 in early summer (lines 363-364, page 13).**

19. Line 365: Figure S6 shows probability of occurrence of compound pollution days under all four types. Please show the situation under type1.

**RESPONSE: The situation under Type 1 and Type 2 are shown in Figure S10.**

Figures and tables:

1. Figure 2 and 3: Please Mark the heavy polluted cases with dark color.

**RESPONSE: Replotted.**

2. Figure 4: Please draw the ridgeline of the subtropical high.

**RESPONSE: Replotted.**

3. Figure 8 and Figure 11: Please compare the difference of daily mean values. An introduction of daily variations makes things difficult to follow.

**RESPONSE: Thanks for your kind suggestion. We have changed Fig. 8 to daily anomalies variation of O$_3$ and PM$_{2.5}$ under impacts of four SWPs over key urban clusters (Fig. 7). Additionally, original Fig. 8 has now been updated to Fig. S5 instead. Original Fig. 11 has now been the revised Fig. 10, with add average value adding to the revised figure.**

[Figure]

**Fig. 7. Daily anomalies variation of O$_3$ and PM$_{2.5}$ under four SWPs in key urban clusters. The black solid line presents the averaged value of each urban cluster.**

**Anonymous Referee #3:**

The authors use the T-mode PCA to objectively classify the summertime synoptic weather pattern across East-Asia and the western Pacific Basin aiming to identify the mode(s) most favorable for compound pollution events across sub-regions in China, specifically for PM2.5 and O3. Many factors governing these events operating across an array of scales are explored. The PCA identified 4 synoptic regimes characterizing the seasonal set up of the 500 hPa WPSH from 2015-2018. An additional large-scale circulation is also at work here, the East-Asian monsoon, which is discussed in context to the WPSH. Additionally, the authors discuss the effects of precipitation frequency and boundary layer characteristics on regulating compound pollution events. Occurrences of pollution are based on Chinese governmental standards.

While this work has great upward potential to be a significant contribution to the community, many revisions are required before publication.

**RESPONSE: We highly appreciate your positive and constructive comments.**

Secs. 1-2 are written quite well and motivate the questions at hand. Beyond that however, I believe that more concrete connections can be and must be made between processes unfolding at different scales (synoptic down to the mesoscale) of motion that lead to Types 1 and 2 dominating the regulation of compound pollution events. For instance, connecting the modulations in the WPSH to changes in favorable PBL conditions and thermal stratification need to be made, in addition to changing precipitation amounts between the types. All of these processes dictate the amount of pollution in the atmosphere at any given time. The final sentence of Sec. 1 states that this manuscript will examine the SWPs responsible for co-occurring pollution events, but the synoptic scale processes have bearing on finer scale processes such as PBL characteristics that are critical for air quality (e.g. inversions associated with tropospheric sinking motion). The authors analyze changes in PBL height between the types and provide loose discussion of their implications for air quality, but further analysis is needed.

**RESPONSE: Thank you for valuable comments. Yes, our motivation is to demonstrate the causes of meteorological processes of the compound $O_3$-$PM_{2.5}$ pollution, as it is believed that the processes should be likely associated with local meteorological conditions (e.g., temperature, wind, humidity, rainfall, and PBL) under the influences of various weather types and modulation of large-scale WPSH movement. In addition, the impacts of PBL characteristics on air quality have been further discussed (lines 414-424 on page 15).**

**Please refer to the following information for more details:**

**"Particularly, Type1 has a significantly warmer temperature over the boundary layer during the compound pollution period of BTH region than that of the clean period. The daytime BLH under compound pollution condition was also higher than that of the clean condition. In addition, there were different directions of prevailing during the two periods, which prevailing winds during the compound pollution period were usually southward and could be driven by air pollutants transported from the southern plains (Fig. 11; see also Miao et al., 2019b, 2020). Co-influencing by the topographical effect of the northern mountainous areas, air pollutants could be trapped in the BTH region. In comparison, although there was southward wind prevailing in the BTH region (Figs. 11 and S11), the rain belt also located in the southern area of BTH might lead to the potential removal of $PM_{2.5}$ (Fig. 9j). Therefore, compound pollution across the BTH region might mainly be due to local emissions of air pollutants."**

Major comments:

1. The abstract needs to be shorted and be more specific.

**RESPONSE: We have reorganized the abstract. Our abstract has been revised as follows:**

**"Surface ozone ($O_3$) pollution during summer (June-August) over eastern China has become more severe, resulting in a co-occurrence of surface $O_3$ and $PM_{2.5}$ (particulate matter with aerodynamic diameter ≤ 2.5 µm in the air) pollution recently. However, the mechanisms regarding how synoptic circulation pattern could influence this compound pollution remains unclear. This study here applied the T-mode principal component analysis (T-PCA) method is used to objectively classify the occurrence of four synoptic weather patterns (SWPs) over eastern China, based on the geopotential heights at 500 hPa during summer (2015-2018). Four SWPs of eastern China are closely related to the western Pacific subtropical high (WPSH), exhibiting, significant intraseasonal and interannual variations. Based on the ground-level air quality and meteorological observations, remarkable spatial and temporal disparities of surface $O_3$ and $PM_{2.5}$ pollution were also found under the impacts of the four SWPs. Particularly, there were two SWPs sensitive to compound pollution (Type 1 and Type 2). Type 1 is characterized by a stable WPSH ridge with axis at about 22°N and the rain belt located in the south of Yangtze River Delta (YRD). High temperature, moderate humidity and low precipitation occurred in the region from BTH to northern YRD (BTH – NYRD), resulting in a co-occurrence of $O_3$ and $PM_{2.5}$ pollution. Additionally, air pollutants can be transported by the prevailing southerly winds from southern plains and accumulated in the southern BTH, resulting in a worsen pollution. Type 2 exhibits a WPSH dominance (the ridge axis ~25°N) and rain belt (over the YRD) in a higher latitude compared with Type 1. High temperature, medium-high humidity and low precipitation over the BTH were the conducive factors related to the occurrence of the compound pollution events under Type 2. Furthermore, low boundary layer height (BLH) and high frequency of light-wind days (FLWD) could create favorable conditions for pollution maintenance. Overall, synoptic weather patterns have played an important role as driving factors of surface $O_3$-$PM_{2.5}$ compound pollution in a regional context. In addition to the impacts of local emissions, our results may provide further insights regarding how regional environmental changes due to co-occurrence of high $PM_{2.5}$ and high $O_3$ level may be driven by the effects of meteorological factors. Overall, our findings demonstrate the important role played by synoptic weather patterns in driving regional surface $O_3$-$PM_{2.5}$ compound pollution, in addition to the large quantities of emissions, and may also provide insights into the regional co-occurring high $PM_{2.5}$ and high $O_3$ level via the effects of certain meteorological factors.".**

2. How do the percentages of the PCs sum to 100%? Shouldn't there be other relevant synoptic patterns than just those 4, meaning that the leading 4 patterns account for most of the synoptic-scale pattern but not all?

**RESPONSE: Thanks for your question. By using T-PCA, users can customize the number of synoptic patterns and determine the domain pattern(s), based on the following information: 1) a distinct direction of the air flow and its related short-term changes, 2) a regime of the pressure field (and vertical movements resulting from the field), 3) particular pattern(s) of front passages, and 4) an inflow of air masses of a particular origin and their related changes.**

**Based on the above method, the similarity of circulation pattern of each day to a particular type expressed by the corresponding loading(s), the greater similarity is expected between the day's type and the pattern(Huth, 1996). Therefore, no matter how many synoptic patterns are predefined, the sum of PCs could be 100%. The final number of patterns is determined by ΔECV, which larger ΔECV means an improved classification performance with stability (Ning et al., 2019). In our study, the highest ΔECV was used to classify to the four patterns. More information has been noted in the supplementary materials.**

**RESPONSE: We appreciate these valuable suggestions. We have now added supplementary information for the vertical motion under impact of four SWPs, and latitude-height cross-sections of mean and anomalous vertical velocity averaged by longitudes over each region under four SWPs. in (Fig. S9). The strong updrafts and positive anomalies, which occurred in some regions (south YRD under Type 1, BTH and GZP under Type 3), is favorable for the formation of precipitation to decrease air pollution. In particular, the downward vertical motion and negative anomalies over BTH under Type 1 and Type 2 are associated with the co-occurrence of O$_3$-PM$_{2.5}$ pollution.**

[Figure]

**Fig. S9. Vertical cross-sections of the means (shading) and anomalies (filled patterns) of vertical velocity (unit: $10^{-2}$ m s$^{-1}$, derived from ERA5 reanalysis data) averaged by longitudes over each region of (a) Type 1, (b) Type 2, (c) Type 3, and (d) Type 4. The dotted and hatched areas represent the negative anomalies less than $-3 \times 10^{-2}$ m s$^{-1}$ and positive anomalies greater than $3 \times 10^{-2}$ m s$^{-1}$, respectively. The gray dashed lines indicate the boundaries of PRD, YRD, GZP, BTH and NEM, and the blank area (23°-27.2°N) is not our study region.**

5. Diffusion of pollutants between the PBL and the free atmosphere is fundamentally related to the turbulent mixing and thermal stratification of the overlying atmosphere. While referenced here, I believe that this is an integral component of this work and must be explicitly addressed across the various subregions. How do the vertical profiles of temperature, moisture, and wind compare across the multiple PCs and subregions? How are these anomalies physically related to the different

synoptic weather pattern differences between the PCs?

RESPONSE: Thank you for your comments. The vertical profiles of temperature, moisture, and wind, as well as their anomalies under sub-regions of each SWP are provided in Fig. S8. Lower WS and its negative anomalies at low level over BTH under Type 1 and Type 2, is not conductive to the diffusion of pollutants. Meanwhile, the moderate RH and its negative anomalies also favor the formation of compound pollution. For GZP, Type 1 and Type 2 correspond to negative anomalies of WS and RH, favoring the occurrence of compound pollution. Note that the probability of compound pollution is relatively small, and it might be related to the local emissions. In other sub-regions, WS mainly affects the diffusion of air pollutants, and precipitation affects the occurrence of ozone and $PM_{2.5}$ pollution to a certain extent.

[Figure]

Fig. R5. The vertical profile of temperature, RH, WS (derived from ERA5 reanalysis data) over subregions under each SWP.

[Figure]

**Fig. S8. The vertical profile of temperature, RH, WS anomalies over subregions under each SWP.**

Other comments:

1. Line 32: "Slight" should be "low"

**RESPONSE: Changed and thanks.**

2. Line 57: insert a "the" before "economy"

**RESPONSE: Inserted and thanks.**

3. Line 72: Change "attached" to "caught"

**RESPONSE: Changed and thanks.**

4. Line 85: "US" should be "United States"

**RESPONSE: Changed and thanks.**

5. Line 105: The Miao et al. findings should be reproducible here but for a multitude of areas. Cross-sections similar to their Figs. 6-7 would work, but regionally averaged vertical profiles would work as well. Vertical profiles of state variables (temperature, stability, vertical velocity, etc.) should be included in this manuscript as these quantities' vertical variation help to significantly modulate PBL and free atmosphere exchange of heat, moisture, pollution, etc. I would also suspect that summertime surface winds would be lower due to more infrequent midlatitude cyclone occurrences, so pollution "pooling" would be more frequent.

RESPONSE: Thank you for your suggestion. Similar analysis as the Figs. 6-7 of Miao et al has been added to the revised manuscript (lines 402-425 on page 15 and Fig. 11). Vertical profiles of state variables are presented in Fig. S8.

Please also find the information as follow:

"In the last section, we have discussed how the SWPs and local meteorological factors modify summer $O_3$ and $PM_{2.5}$ pollution. However, how does the boundary layer structure interact with the co-occurrence of $O_3$-$PM_{2.5}$ pollution? In order to address this question, we conducted a further analysis. As mentioned, co-occurrence of $O_3$ and $PM_{2.5}$ pollution were mainly found in the BTH – NYRD under Type 1 and over BTH region under Type 2. Lower WS and its negative anomalies at lower boundary layer over BTH– NYRD under Type 1 and over BTH under Type 2 may not enhance the diffusion of air pollutants (Fig. S8). In contrast, moderate RH and its negative anomalies might favor the formation of compound pollution. Downward vertical motion and negative anomalies could also stabilize the atmospheric characteristics of boundary layer (Fig. S9). Furthermore, we summarized boundary layer structure, precipitation, and ground-level wind flow across the BTH region. Based on the characteristics, we separately defined Type 1 and Type 2 into clean (both concentrations of the $O_3$ and $PM_{2.5}$ are less than polluted level) and compound pollution periods (Figs. 11 and S10-S11). Particularly, Type1 has a significantly warmer temperature over the boundary layer during the compound pollution period of BTH region than that of the clean period. The daytime BLH under compound pollution condition was also higher than that of the clean condition. In addition, there were different directions of prevailing during the two periods, which prevailing winds during the compound pollution period were usually southward and could be driven by air pollutants transported from the southern plains (Fig. 11; see also Miao et al., 2019b, 2020). Co-influencing by the topographical effect of the northern mountainous areas, air pollutants could be trapped in the BTH region. In comparison, although there was southward wind prevailing in the BTH region (Figs. 11 and S11), the rain belt also located in the southern area of BTH might lead to the potential removal of $PM_{2.5}$ (Fig. 9j). Therefore, compound pollution across the BTH region might mainly be due to local emissions of air pollutants."

[Figure]

**Fig. 11. The daily variation of horizonal wind, potential temperature (PT) and BLH of boundary layer in the BTH under clean and compound pollution period of Type 1 and Type 2 (a, b, e, f). The vertical cross-section of u-wind, w-wind and PT for the same situation of BTH (c, d, g, h). The w-wind is multiplied by 100 when used. The data has been derived from ERA5 reanalysis data.**

6. Lines 146-147: Subregions should be labeled in a figure to orient the reader. This can be done in panel (a) of Fig. 1.

**RESPONSE: Revised.**

7. Fig 1.: There is no "red box"? If there is, it is not clear

**RESPONSE: We have changed to the "black box".**

8. Figs. 2 and 3: Please change the color of "heavily polluted" regions to something other than turquoise. It can easily be misinterpreted as a "good" category

**RESPONSE: Changed.**

9. Line 200: How is "pollution day defined" for O3? By the thresholds laid out earlier (160 ug/m3 threshold)? Also, what constitutes "moderate" pollution? Same question applies for PM2.5.

**RESPONSE: The pollution levels of $O_3$ and $PM_{2.5}$ over each key area were verified according to the limit of air pollutant concentration, based on the Technical Regulation on Ambient Air Quality Index (on trial) (HJ633-2012) issued by the Ministry of Ecology and Environment of the People's Republic of China Specific standard limits are now shown in Table S1 of the supplementary materials.**

**Table S1. Thresholds for each pollution level of $PM_{2.5}$ and $O_3$-8h.**

| AQI | Pollution level | $PM_{2.5}(\mu g\ m^{-3})$ | $O_3$-8h ($\mu g\ m^{-3}$) |
|---|---|---|---|
| 0~50 | Good | 0~35 | 0~100 |
| 51~100 | Moderate | 36~75 | 101~160 |
| 101~150 | Lightly polluted | 76~115 | 161~215 |
| 151~200 | Moderately polluted | 116~150 | 216~265 |
| 201~300 | Heavily polluted | 151~250 | 266~800 |

10. Line 219-220: This low-level transport feature and its variation with PC is not shown in any figure but is referenced. I believe that at least one figure should show this feature since it is being discussed in forthcoming results

**RESPONSE: The low-level transport feature is shown in Fig. 11.**

11. Line 226: How can you infer that water vapor is being transported into the YRD regions? The 850-hPa flow vectors are at best directed parallel to the YRD coastline. Otherwise they are directed offshore. Additionally, inferences about moisture transport should be made by wind/water vapor overlays or by integrated vapor transport/moisture convergence analysis (see Rahimi et al. 2018), which this figure does not have.

**RESPONSE: Thank you for your valuable suggestion. We added the water vapor flux (WVF) in Fig. 4. WVF denotes the direction and magnitude of atmospheric moisture transport, which is simplified as : WVF = $V*q$/g, where q is specific humidity, g is the gravitational acceleration (= 9.8 g/m²), and V is the horizontal wind. It can be seen that the southwesterly flow transports sufficient water vapor to the YRD region.**

[Figure]

**Fig. 4. 850-hPa water vapor flux (WVF =V\*q/g, q is specific humidity, g is gravitational acceleration, V is horizonal wind; vectors; see scale arrow at the bottom right in units of 5 g cm⁻¹hPa⁻¹s⁻¹) and 500-hPa GH (contours; see scale bar at bottom in units of gpm) patterns based on objective classification (see text for details). White box area is for the area of eastern China, the number at the upper right corner of each panel indicates the frequency of the occurrence of each pattern type, the black line of each panel presents the ridge axis of the WPSH.**

12. Line 227: Flow shifting from southwesterly to westerly with northward extent is anticyclonic, which we see in the figure. At the same time, we see a cyclonic sped shear maximum, so it is impossible without quantifying the vorticity explicitly to say if this is anticyclonic or cyclonic. Please remove "cyclonic."

**RESPONSE: Thanks for your comment. Removed.**

13. Line 229: Is it the WPSH retreating or advancing? Its axis appears to shift north slightly. . .

**RESPONSE: The WPSH shifts northward slightly from Type 1 to Type 2; and it retreat southeastward from Type 3 to Type 2. Thanks for the suggestion and we have now revised the content (Lines 274-275 on page 10).**

14. Line 230: Consider deleting, "and the GH over northern China at 500 hPa is higher compared with Type 1 (Fig. 4b)." The change in the magnitudes of GH are not terribly important; it is the change in their gradients that regulates the winds in each PC. Line 231: The only Type 2 onshore flow (at 850 hPa) I see is around 120E by 30N, which lies directly west of the Type 2 GH maximum. This is an example of how you can use certain language relating flow properties to GH anomalies

for specific PCs. Currently the authors state, ". . .southerly wind blowing from the ocean to the continent lies in front of the bottom of the high pressure,. . .", which is very unclear. More generally, I recommend the authors refrain from using "top" and "bottom" unless they are referring to the vertical axis (i.e. up and down).

**RESPONSE: Thank you for your comment. The related sentence in Line 230 has been deleted. We also changed ". . .southerly wind blowing from the ocean to the continent lies in front of the bottom of the high pressure,. . ." to "The southerly wind from the ocean could interact with northern periphery of the WPSH".**

15. Line 233: "and the rain belt moves northwards to the east of the YRD region." Are the authors referring to the belt as it compares to other PCs? If so, the different PCs may be compared to one another, but it is not guaranteed that any type will necessarily evolve from another type. Please clarify and reword throughout the text.

**RESPONSE: Thank you for your suggestion. We have reworded the descriptions for the rain belt and plotted the rain belt in Fig. 8 (i-l). The discussion about rain belt is moved to Sec.3.4 as follows:**

**"Type 1 is characterized by humid condition in the southern area and dry condition in the northern region owing to an extensive southwestern flow of the WPSH, resulting in a rain belt found in southeastern coastal area such as PRD and YRD regions. Type 2 is associated with meridional flow and dry and wet anomalies in northern China, resulting in a rain band locating at the central areas of between BTH and YRD regions due to the northern advance of the WPSH compared with Type 1. Furthermore, there is a greater RH for most of the study sites under Type 3 and Type 4, possibly a result of the shifted rain belt in the BTH and NEM regions under Type 3 once the northern boundary of the WPSH reaching at 37.5°N, and an occurrence of heavy precipitation across the western PRD region as well as central areas of between BTH and YRD regions under Type 4 (Fig. S6)".**

16. Lines 237-238: ". . .which implies that the rainy season in the YRD region ends in midsummer and the weather becomes hot and dry." How is this implied? 850-hPa flow is onshore beneath the western edge of the 500 hPa monomer of the WPSH. This linkage is not implicit and should be explained. Moreover, references made to shifts in precipitation need to be explicitly shown if they are going to be frequently referred in the text.

**RESPONSE: We appreciate your advice. We have revised this sentence as follow: "This has led to a condition completely controlled by the monomer of the WPSH over the YRD region, resulting a hot and dry weather at the end the rainy season at the beginning of mid-summer.". As for rainy season, we have determined the location of the rain belt based on the amount of precipitation and PF under each SWP, which can be seen at Fig.8(i-l).**

[Figure]

**Fig. 8. Same as Fig. 6 but for Tmax (a–d), RH (e–h), and PF (i–l). The black solid line presents the rain belt of each SWP.**

17. Lines 239-240: "continues to extend westwards and shift northwards," shifts westward and northward compared to PC3. Again, please indicate it's the synoptic pattern's position more explicitly. Something like, "In Fig. 4d, the monomer is located north and west of the feature in Fig. 4c". The word accordingly relates this sentence to the previous one, but it isn't clear what that

linkage actually is. Also, please explain the linkage or remove the word "accordingly."

**RESPONSE: This sentence has been changed as follow: "Figure. 4d indicated that the location of WPSH monomer was more western and northern with respect to other SWPs, controlling the northern China for a long time; the western ridge point was around 95°E and the northern boundary was around 40°N.".**

18. Line 241: Heat wave? How is PC4 related to a heat wave? Where is this shown in the figures or analyses?

**RESPONSE: Sorry for the confusion. The words related to "heat wave" have now been removed from this sentence.**

19. Figs 3-4: How are levels of air quality defined? Are they arbitrary? If so, then a brief justification is required. If they are a community standard, then a source is needed.

**RESPONSE: The pollution levels of $O_3$ and $PM_{2.5}$ over each key area were verified according to the limit of air pollutant concentration, based on the Technical Regulation on Ambient Air Quality Index (on trial) (HJ633-2012) issued by the Ministry of Ecology and Environment of the People's Republic of China Specific standard limits are now shown in Table S1 of the supplementary materials.**

20. Fig. 4 shows the PCs of the synoptic weather pattern and associated percentages of occurrence for the study period.

**RESPONSE: Yes.**

21. Fig. 5: 2017 is labeled twice. Should the second instance be 2018?

**RESPONSE: Revised and thanks.**

22. Line 263: Any hypothesis for why the lowest MDA8O3 occurs for PC3? Could it be related to the synoptic pattern of Fig. 4 and associated precipitation (not shown)?

**RESPONSE: The analysis of light $O_3$ pollution under Type 3 is revised as follow (lines 448-455 on pages 16-17):**

**"High temperature, low humidity and few precipitations over the YRD region tend to generate a large amount of $O_3$, while the positive BLH and negative FLWD anomalies are unfavorable to $O_3$ accumulation. On the other hand, summer typhoon activities might weaken the WPSH intensity over the YRD region, leading to the eastward retreat and northward shift of the WPSH. As a result, high WS across coastal areas could ease the ground-level $O_3$ pollution (Shu et al., 2016). For the BTH and PRD regions, high PF tends to suppress $O_3$ production. Only 6.8% of the compound pollution occurrence days at all sites occurred under Type 3, in accordant with the light $O_3$-only pollution over the areas of the BTH, YRD and PRD (Fig. 12)."**

23. Line 279: Delete "in the eastern region"

**RESPONSE: Deleted.**

24. Fig. 8, Lin3 285: What constitutes "serious?" Perhaps it would be good to plot the pollution threshold values here for O3 and PM2.5. Plotting these curves (they would be straight lines) would help the reader to identify how bad (or good) the air quality actually is in terms of PM2.5 and O3. The authors discuss "over-standard" rates, so a threshold must have been used (plot it). I believe these values are 160 and 75 ug/m3 for O3 and PM2.5, respectively. . .

**RESPONSE: We appreciate your comment. The threshold values of $O_3$ and $PM_{2.5}$ for "over-standard" refer to the 24-h concentrations, and we explained the threshold values for $O_3$ and $PM_{2.5}$ in the Data and methods as "Based on the Ambient Air Quality Standards (GB3095-2012) issued by the Ministry of Ecology and Environment of the People's Republic of China,**

**O₃ (PM₂.₅) pollution occurs when the MDA8 O₃ (PM₂.₅ 24-h) concentration exceeds 160 (75) µg m⁻³." In order to more clearly compare the concentration differences under different SWPs in various regions, we have changed the Fig. 8 to daily anomalies variation.**

[Figure]

**Fig. 7. Daily anomalies variation of O₃ and PM₂.₅ under four SWPs in key urban clusters. The black solid line presents the averaged value of each urban cluster.**

25. Line 286: For (2), over-standard rates are not plotted – concentrations are. Please clarify. If the authors are suggesting that O3 an PM2.5 concentrations are similar between PCs, then please reword.

**RESPONSE: The over-standard rates are shown in Table 1. Reworded it to "in the PRD region, the over-standard rates and concentrations of O₃ and PM₂.₅ is similar under four SWPs".**

26. Line 287: For (3), it looks like Type 4 is leading, not Type 1, for O3 concentrations from 0900-1500. Since this is when concentrations are largest, the "Type 1 > Type 4 > . . ." may mischaracterize your argument.

**RESPONSE: The O₃ pollution over-limit ratio is calculated via stations* days, it presents as Type 1 > Type 4 > Type 2 > Type 3" in the YRD. But it is true that Type 4 is leading, not Type 1, for O3 concentrations from 0900-1500, this is because the daily variation is counted by regional average concentrations.**

27. Line 302: "Activities"? Do the authors mean "modulations"? This is unclear..

**RESPONSE: Thank you for the suggestion. We have now used "modulations" (line 87, page4) as suggested.**

28. Line 308: "Makes summer always hot and moist" grammar needs revisions

**RESPONSE: Revised. We changed it to "induces a hot and humid condition over the summer".**

29. Line 316: "presents negative" should be followed by "(Fig. 9a)" to guide the reader. Also, why are Tmax anomalies negative under this PC?

**RESPONSE: We have added the tags to guide the reader. The reason of negative Tmax anomalies under Type 1 is that Type 1 is always occurring early summer.**

30. Lines 312-321: Precipitation is integral to the lifecycle of PM2.5 and O3. The linkages between the precipitation anomalies and Fig. 4 should be explicitly discussed. I believe the authors attempt to do this in Sec. 3.2, but that discussion is more appropriate here.

**RESPONSE: Thanks for your advice. We have moved the discussion of precipitation to subsection 3.4. Please refer to lines 352-361 on page 13: "Type 1 is characterized by humid**

**condition in the southern area and dry condition in the northern region owing to an extensive southwestern flow of the WPSH, resulting in a rain belt found in southeastern coastal area such as PRD and YRD regions. Type 2 is associated with meridional flow and dry and wet anomalies in northern China, resulting in a rain band locating at the central areas of between BTH and YRD regions due to the northern advance of the WPSH compared with Type 1. Furthermore, there is a greater RH for most of the study sites under Type 3 and Type 4, possibly a result of the shifted rain belt in the BTH and NEM regions under Type 3 once the northern boundary of the WPSH reaching at 37.5°N, and an occurrence of heavy precipitation across the western PRD region as well as central areas of between BTH and YRD regions under Type 4 (Fig. S6).".**

31. Lines 328-331: This sentence is unclear and should be revised. Also, there is an instance here where the authors use an acronym in one part of the sentence but not the other. Please be consistent. Also, how do negative FLWD anomalies result in a deeper PBL?

**RESPONSE: Thank you for your comment. It has been revised to "As can be seen from Fig. 9, when the BLH has a positive anomaly, on the contrary FLWD has a negative anomaly (e.g., BTH in Type 1), which indicates the higher BLH, the lower FLWD, the more conducive to the diffusion of pollutants, otherwise, lower BLH and higher FLWD (BTH in Type 2) do not support the diffusion.". The averaged WS would be higher when negative FLWD anomalies, usually the BLH is higher in this case.**

32. Sec. 4.1, P3: I believe that stability should be discussed here in addition to a more detailed discussion of precipitation "anomalies" associated with each PC. Thermal stratification of the PBL will dictate the mixing depth of the PBL and thus regulate the pooling of these aerosol/pollution plumes. Looking at the correlation between Tmax, PF, FLWD, etc. is not enough.

**RESPONSE: Thank you for your suggestion. The stability discussion is linked to thermal stratification of the PBL in Sec. 4. Please refer to the response to major comments 3 of referee #1 and major comments 4-5 of yours.**

33. Lines 346-349: Here is a wonderful chance to discuss what is special about PC4 on a synoptic level. Why is PC4 leading to the largest O3 events synoptically? Do these same conditions favor the co-occurrence of O3 and PM2.5?

**RESPONSE: We appreciate your suggestions. We have discussed these questions in the Discussion as follows:**

**"High temperatures, medium-high humidity and few precipitations in the GZP and PRD regions can cause $O_3$-$PM_{2.5}$ compound pollution, but $PM_{2.5}$ pollution in the both regions is are not heavy, which is possibly in relation to local lower pollutant emissions. The probability of compound pollution occurrence under Type 4 is about 5.1%. Under the control of the WPSH, there are strong photochemical reactions at high temperatures and little rainfall in some eastern region (such as North BTH, YRD), which is also conducive to $O_3$ generation (Fig. 12). Meanwhile, relative to Type 1, $O_3$ pollution is lighter in the BTH, due to the differences of RH, BLH and FLWD."**

34. What is the difference between the Yangtze River and the YRD? These should be labeled on a map for readers. . .

**RESPONSE: It should be "YRD" there.**

35. It seems as though there is a window of moisture availability that is large enough to allow

hygroscopic growth of PM2.5 but is sufficiently small to allow for the important photochemical processes that regulate O3. It would seem to me that identifying this moisture window, as well as its sensitivity to PCs, would be a very significant contribution and I recommend that it be studied further to more precisely identify the PCs responsible for co-occurring O3/PM2.5 events. Identification of this moisture window can be based on an optimal relative humidity for compound pollution events too. This window can change by region and PC type.

**RESPONSE: Thank you for your valuable comment. We selected the BTH region, an area with high frequency of compound pollution, to analyze the RH condition under four period (compound pollution, clean, O$_3$-only, PM$_{2.5}$-only). Indeed, there is a moisture window here, higher RH would restrain the production of O$_3$, and lower RH would not favor hygroscopic growth of PM$_{2.5}$. The moderate RH is more conducive to the formation of compound pollution.**

[Figure]

**Fig. S14. Box-Whiskers for the RH under compound pollution, clean, O$_3$-only, PM$_{2.5}$-only period. In the Box-Whiskers plot, the central box represents the values from the lower to upper quartile (25th to 75th percentile). The vertical line extends from the maximum to the minimum value. The middle solid line represents the median, and the red plus represents the outlier.**

36. Line 368: Strengthens compared to Type 1? Type 2's trough does not necessarily strengthen from the Type 1 pattern. Please reword.

**RESPONSE: Thank you for your suggestion. We have reworded it to "As the northern advance of WPSH from Type 1 or the retreat from Type 3 or Type 4, and the northern region is still controlled by the westerly zone".**

37. Lines 357-387: The authors give the percentage of days with compound pollution for types 1 and 2. However, this does not elucidate which type is more efficient at producing compound pollution. The authors should include the percentages of compound pollution days for types 3 and 4. From the results here, I'd suspect that types 1 and 2 are the most efficient compound pollution setups, but this can be confirmed by including the percentages as for types 1 and 2.

**RESPONSE: Thanks for your suggestion. The probability of compound pollution occurrence**

**under Type 3 and Type 4 are 6.8% and 5.1%, which are added in lines 453 and 459.**

38. Lines 396-397: These percentages need to be presented for Types 3 and 4 as higher percentages may indicate that PCs 3 and 4 are more efficient setups for co-occurring pollution events, even if the PCs occur less frequently.

**RESPONSE: Thanks and added.**

39. Line 398: "line" should be "axis"

**RESPONSE: Revised and thanks.**

40. Line 400: Again, what is "Meiyu" season for non-local readers?

**RESPONSE: The explanation about "Meiyu" is added to lines 270-274 on page 10: "These conditions were also associated with high temperature and humidity during the summer with Meiyu season, which Meiyu season is a climate phenomenon with continuous cloudy and rainy days generally occurring during June and July every year in the middle and lower reaches of Yangtze river, Taiwan of China, central/southern Japan, and southern Korea".**

41. Lines 400-401: How do higher temperatures suppress O3 production? I would suspect that the higher relative humidifies are primarily responsible. . ..

**RESPONSE: Yes, you are right. We have revised it.**

42. Line 403: Is the low pressure trough at the surface or at 500 hPa?

**RESPONSE: It is referring to the condition at 500 hPa. We have added related information to Lines 478, page 17.**

43. Lines 402-404: Again, this "small amount of water vapor transport" suggests that there is a nominal vapor pressure deficit conducive to compound pollution events. In an environment of appropriate stability and PBL characteristics, compound pollution may be especially severe.

**RESPONSE: Thank you for your valuable suggestion. We have revised it as "The hygroscopic growth of PM$_{2.5}$ occurs in the corresponding area in front of the trough with a small amount of water vapor transported by the WPSH. Particularly, the prevailing southerly winds in the boundary layer can transport the pollutants emitted from southern cities to the BTH region, and the atmospheric stratification is stable when the air mass is sinking. Thus, the compound pollution can be severe. In general, the synoptic circulation might be responsible for the concentration of pollutants under this SWP".**

44. Lines 407-408: It appears that the WPSH only shifts north in your objective PC analysis, not southwards and eastwards. . .can the authors clarify?

**RESPONSE: From Type 1 or Type 2 to Type 3 or Type 4 presents the shift north of WPSH in early summer, in the contrary, from Type 3 or Type 4 to Type 2 or Type 1 presents southwards and eastwards retreat. We have added the location of the WPSH (Table S1) and re-described the motion of the WPSH. Please refer to lines 266-285 on pages 10-11, lines 474-475 and 486-489 on pages 17-18.**

45. Line 411: Why does water vapor lead to a sink of O3? Water vapor by itself cannot remove O3 from the atmosphere or prevent its formation. Are the authors referring to the supersaturation, dew point depression, etc.?

**RESPONSE: Sorry for the confusion. We did not refer to the supersaturation or dew point depression. Instead, we are referring to water vapor flux at 850 hPa (Fig. 4). This is based on the following context: as water vapor can absorb part of the short-wave solar radiation, and this can weaken the photochemical reaction and reduce ozone production. The details have now been revised in the manuscript (Lines 489-490 of Pages 18).**

*Marked-up manuscript with track changes*

[revised manuscript text omitted]
~~The cost733software includes several modules for classification, evaluation and comparison. The T-PCA classification on cost733 performs spatial standardization on data, splits data into 10 subsets, achieves the principal components (PCs) according to singular value decomposition, then applies oblique rotation to PCs, calculates the PC score of each subset, finally compares 10 subsets and selects the most consistent one with other classification (Miao et al. 2017).More detailed information about the T-PCA method can be found in Miao et al. (2017).~~ To assess the performance of synoptic classification and determine the number of classes, the explained cluster variance (ECV) is selected in this study (Hoffmann & SchlüNzen, 2013; Ning et al., 2019; Philipp et al., 2014). The detailed information about the ECV is provided in the supplementary document.

Based on the Ambient Air Quality Standards (GB3095-2012)

[revised manuscript text omitted]

330 and climate of East Asia (Lu, 2002), owing to its various location, shape and intensity (Ding, 1994). A low-level southerly monsoon formed at the periphery of the WPSH can transport warm and humid air from the ocean to East Asia, which might also be responsible for the asymmetric spatial distribution in response to an enhanced WPSH for ground-level $O_3$, i.e. a decrease in southern China but an increase over northern China (Zhao & Wang, 2017).

335 Therefore, we used the T-PCA method to objectively classify the weather circulation of the 500-hPa GH field in the summers of 2015–2018, and finally obtained four SWPs related to the movement and development of the WPSH. The location of western ridge point and northern boundary of the WPSH at 500 hPa in Type 1 is

aboutround 120°E and 30°N, respectively, (Fig. 4a and Table S2). as shown in Fig. 4a,The southwestern flow of theis WPSH could transports water vapor into the YRD region, and resulting in a southwestwardthe prevailing windsouthwesterly in across the YRD region and westward flow from the north of the WPSH forming a cyclonic convergence area at 850 hPa,. These conditions were also associated with high temperature and high humidity during the summer with Meiyu season, which Meiyu season is a climate phenomenon with continuous cloudy and rainy days generally occurring during June and July every year in the middle and lower reaches of Yangtze river, Taiwan of China, central/southern Japan, and southern Korea). For Type 2, it is noticed that the westerly trough could deepens accompanying as the WPSH shifts northward slightly from Type1 or retreats southeast from Type 3 (or southward) advance (retreat) of WPSH, and the GH over northern China at 500 hPa is higher compared with Type 1 (Fig. 4b). The southerly wind blowing from the ocean to the continent lies in front of thecould interact with northern peripherybottom of the high pressureWPSH,. As a result, the sea-land interaction could interact withaffecting the southeastern region across China, while northern China iscould be mainly controlled by the westerly trough, and the rain belt moves northwards to the east of the YRD region. In compared with Type 2, UnderType 3, presents the boundary of WPSH shifts further northin a higher latitude, with a westward extension compared with Type 2 (Fig. 4c), and disintegrates disintegrating a closed high-pressure monomer along the eastern coast of China, while and the main body of the WPSH remains onover the ocean (Figs. 4c and S4). This has led to a condition completely controlled by the monomer of the WPSH over the YRD region, resulting a hot and dry weather at the end the rainy season at the beginning of mid-summer.In this case, the YRD region is completely controlled by the monomer of the WPSH, the region ends the rainy season and enters midsummer, becoming hot and dry., which implies that the rainy season in the YRD region ends in midsummer and the weather becomes hot and dry.At the same time, the rain belt moves gradually northwards to the BTH and NEM regions. According to InFigure. 4d, indicated that the locationmonomer of the WPSH monomer iwas more western and northern with respect to other SWPslocated the north and west of the feature in Fig.4c, under Type 4 continues to extend westwards and shift northwards, controlling the northern China for a long time,; the western ridge point iwas aboutaround 95°E and the northern boundary iwas aboutaround 40°N, which location of the WPSH is more west and north than other SWPsand with

.

[revised manuscript text omitted]

marked with a '+' indicates the Analysis of Variance passes the significance level of 0.05.

[Figure]

**Fig. 7. The PM₂.₅ anomaly under four SWPs, where the sites marked with a '+' indicates the**

990            **Analysis of Variance passes the significance level of 0.05.**

[Figure]

**Fig. 87. Daily variation of O₃ and PM₂.₅ anomalies under four SWPs in key urban clusters.**

995

[Figure]

[Figure]

**Fig. 8. Same as Fig. 6 but for Tmax (a–d), RH (e–h), and PF (i–l). The black solid line presents the rain belt of each SWP.**

[Figure]

**Fig.** 9. Same as Fig. 6 but for BLH at 14:00 BJT (a–d) and FLWD (e–h).

1005

[Figure]

1010        **Fig. 1110. Daily variation of NO$_2$ (a-e) and Ox (f-j) under four SWPs in key urban clusters.NO$_2$ and O$_3$/NO$_2$ daily variation under four SWPs in key urban clusters.**

[Figure]

**Fig. 11. The daily variation of horizonal wind, potential temperature (PT) and BLH of boundary layer in the BTH under clean and compound pollution period of Type 1 and Type 2 (a, b, e, f). The vertical cross-section of u-wind, w-wind and PT for the same situation of BTH (c, d, g, h). The w-wind is multiplied by 100 when used. The data has been derived from ERA5 reanalysis.**

[Figure]

[Figure]

**Fig. 12. Schematic diagrams describing the relationships between the WPSH, four SWPs and**

1025 **summertime O₃ and PM₂.₅ pollution in various regions.**

---

## Referee Report (RR1)

The manuscript has been improved, but still need further revision to meet the standard of ACP. Some of my concerns were not well addressed.

1. "The number of synoptic patterns (k) is optimized when the $\Delta$ECV is at the highest value, which suggests that the performance of classification has been improved substantially and with stability." The authors presented the explained cluster variances from 4 types to 15 types. How about 2 or 3 types? The results can have a higher increment of the ECV? The highest value of $\Delta$ECV is no guarantee of reliable classifications. More in-depth analysis and discussions on the 4-type classification results may be added, as well as its uncertainties and limitations. A specific synoptic pattern can be caused by the seasonal movement of WPSH or the quick pass of a typhoon, which can lead to different atmospheric processes (e.g. precipitation, LLJ, large-scale subsidence) and pollution levels.

2. The detailed descriptions of typhoon-case (Fig. R1 and R2) can be added in the revised manuscript to help readers to understand the sharp movement of WPSH.

3. The ERA5 data were used in this study, but not described in the manuscript. Why not classify the 500-hPa fields of ERA-5, and then carefully analyzed the PBL and precipitation based on the hourly ERA-5 data. How about the consistencies/differences between the ERA-5 data and NCEP data.

4. How many sounding profiles at 08, 14 and 20 BJT were used in this study for each studied city? How to use 08 and 20 LT soundings to estimate the afternoon BLH? Please clarify. In summer, the relationships between BLH and concurring/compound pollution in East China are quite complicated due to the transport of precursors (https://doi.org/10.1016/j.envpol.2020.115775). More in-depth analysis/discussion on the PBL-pollution linkage and transport of precursors in East China must be added.

5. Please carefully check the cited papers, some were not properly. For example, the BLH estimation method was actually from the study of Seidel et al. (2012, https://doi.org/10.1029/2012JD018143).

---

## Referee Report (RR2)

**Review of acp-2020-596: Synoptic drivers of co-occurring summertime ozone and PM$_{2.5}$ pollution in eastern China**

The authors use the T-mode PCA to objectively classify the summertime synoptic weather pattern across East-Asia and the western Pacific Basin aiming to identify the mode(s) most favorable for compound pollution events across sub-regions in China, specifically for PM2.5 and O3. Many factors governing these events operating across an array of scales are explored. The PCA identified 4 synoptic regimes characterizing the seasonal set up of the 500 hPa WPSH from 2015-2018. An additional large-scale circulation is also at work here, the East-Asian monsoon, which is discussed in context to the WPSH. Additionally, the authors discuss the effects of precipitation frequency and boundary layer characteristics on regulating compound pollution events. Occurrences of pollution are based on Chinese governmental standards.

The authors present a much-improved manuscript. The authors now show a clear connection between different synoptic modes and compound pollution events across different sub-regions in eastern Asia. The authors link the favorable synoptic modes to favorably meteorological conditions in Tmax, wind, stability, and more.

I believe that this paper will be ready for publication once its grammar has been improved. Thus, I recommend major revisions at this time, but I must emphasize that the authors should be proud of the improvements they have made to this manuscript. There is a strong message developing. With the proper grammatical improvements, this will be a significant contribution to the literature.

1. The abstract can and should be shortened considerably. The authors have identified two preferred SWPs conducive for compound pollution events and then provide many details. The details can be left to the main text and omitted from the abstract. Furthermore, the abstract should be in the same tense. Currently, there is a mix of past tense and present tense expository.
2. Line 79: What does "gradually been prominent" mean? Do the authors mean that O3 pollution in summer has increased in recent years?
3. Line 90: WS, please define as wind speed. I do not it is defined previously in the main text.
4. Line 105: When referring to previous studies, present material in the past tense, but the rest of the paper should be written in the present tense.
5. Line 116: "anomalies" should be "anomaly"
6. Line 129: Are the winds southerly or northwesterly?
7. Line 137: Delete "simulation"
8. Line 146: Should be "pollutants"
9. Line 192: "consists" should be "consisting"
10. Line 193: "pattern" should be "patterns"
11. Lines 194-197: This sentence needs to be reworked grammar-wise.
12. Line 205: Are the authors counting days as O3 and PM2.5 days when > 50% of the sites exceed the aforementioned thresholds? If so, the grammar here needs to be reworked.

13. Line 231: "The" should be "the"
14. Line 232: New sentence should begin at ", as aresult"
15. Lines 233-237: Are the authors referring to the total days in the 2015-2018 period?
16. Line 237: This sentence is repeated
17. Line 255: Wait – are Figs. 2-3 composited only on days characterized by SWPs 1-4? I thought these for all days? If for all days, delete "days for four SWPs"
18. Lines 247-248: Change the wording of this sentence."These results indicate that, despite PM2.5 reductions, compound pollution events deserve public attention." Delete the following sentence.
19. Lines 261-263: This sentence needs to be reworded from "which might…" onwards
20. Line 273: Change "in" to "across"
21. Lines 275-278: This sentence is hard to follow and needs to be reworked. For example, how can the sea-land interaction interact with the southeastern region across China? I think the authors can just explain the different spatial configurations of the different modes of the WPSH and leave discussion for later on when discussing the compound pollution event conditions
22. Line 535: "locating" should be located
23. Line 415: Prevailing…."winds?"
24. Lines 418-420: Were the prevailing winds driving pollution transport from the southern plains? This sentence needs to be reworked grammatically.
25. Fig. 11: Panels are uneven. Please replot
26. Lines 427-463: These points can be shortened, and the grammar needs to be revised. A lot of the discussion for this passage was made in previous sections.

---

## Author Response (AR2)

**Reponses to referee(s) comments**

**Dear Editor,**

**Thank you for your efforts for handling our manuscript. We appreciate to receive the useful comments from all referees. These comments are very constructive, and we have now further revised our manuscript in light of all referees' comments. Based on the helpful suggestions from all reviewers, we believe that we should have addressed questions and concerns from all referees appropriately, and adequately. Please find our point-by-point responses below.**

**Anonymous Referee #1:**

The manuscript has been improved, but still need further revision to meet the standard of ACP. Some of my concerns were not well addressed.

**Response: Thank you for your valuable time to review this manuscript. We have further revised the manuscript based on your constructive comments.**

1. "The number of synoptic patterns (k) is optimized when the ΔECV is at the highest value, which suggests that the performance of classification has been improved substantially and with stability." The authors presented the explained cluster variances from 4 types to 15 types. How about 2 or 3 types? The results can have a higher increment of the ECV? The highest value of ΔECV is no guarantee of reliable classifications. More in-depth analysis and discussions on the 4-type classification results may be added, as well as its uncertainties and limitations. A specific synoptic pattern can be caused by the seasonal movement of WPSH or the quick pass of a typhoon, which can lead to different atmospheric processes (e.g. precipitation, LLJ, large-scale subsidence) and pollution levels.

**RESPONSE: Many thanks for your valuable suggestions. An important criterion to determine the number of SWPs is to ensure that the differences between different synoptic patterns are the largest, while the differences within the same synoptic pattern is the smallest. ECV is usually recommended as an indication, as a greater ECV value often corresponds to a better performance of the synoptic pattern classification (Hoffmann and Heinke SchlüNzen, 2013). The highest value of ΔECV means that the performance in the synoptic pattern classification is improved substantially (Ning et al., 2019). Therefore, both higher ECV and ΔECV values were considered in our study. We found the small value of ECV when the number of SWPs was two or three, indicating greater differences within the same synoptic pattern. The ECV value showed the highest increase when the number of SWPs was four, which means the differences within the same synoptic pattern was significantly improved (Fig. S1). Therefore, four SWPs were finally selected in our study. We have added the above analysis in the text S1 of supplementary material. We have added more discussions about classification results, uncertainties and limitations at lines 471–480 on page 17. Please also see as follow:**

**"It is important to note that our work contains a few limitations and uncertainties. Although T-PCA, an objective classification method, was chosen in this study, there were still some subjective decisions made, e.g., the number of SWPs (Huth et al., 2008). In the present work, we selected four SWPs based on both the larger ECV and greater ΔECV to furthest reduce the subjective impact. Nevertheless, at a large scale, the present four SWPs were closely associated with intraseasonal movements of the WPSH, because the WPSH is one of the most**

important components of the present large-scale SWPs in summertime (Zhao and Wang, 2017). In addition, note that short-term disturbances induced by typhoons with specific pattern were not excluded. The quick passage of a typhoon in summer could lead to various atmospheric processes (e.g., precipitation, large-scale subsidence) and pollution levels (Deng et al., 2019), which should be explored in future work."

2. The detailed descriptions of typhoon-case (Fig. R1 and R2) can be added in the revised manuscript to help readers to understand the sharp movement of WPSH.

RESPONSE: We appreciate your kind suggestion. We have added these at lines 295–300 on page 11. Please also see as follow:

"For instance, tropical storm NEPARTAK generated at 0000 UTC (0800 BJT) 3 July 2016 over the western North Pacific and upgraded to a super typhoon at 1200 UTC (2000 BJT) 5 July 2016 (Fig. S5; see also Su et al., 2017). Due to the rapid movement of NEPARTAK to the northwest, the WPSH quickly decomposed a monomer and moved north. With the strengthening and landing of the typhoon, the monomer gradually collapsed. The SWP also underwent a transition from Type 2 to Type 4, and then to Type 1 (Figs. 4 and S5).".

Reference:

Su, H., Qian, C., Gu, H. and Wang, Q.: The Impact of Tropical Cyclones on China in 2016, Trop. Cyclone Res. Rev., 5(1–2), 1–11, doi:10.6057/2017TCRRh1.01, 2017

3. The ERA5 data were used in this study, but not described in the manuscript. Why not classify the 500-hPa fields of ERA-5, and then carefully analyzed the PBL and precipitation based on the hourly ERA-5 data. How about the consistencies/differences between the ERA-5 data and NCEP data.

Response: Sorry for our negligence. The description of ERA5 data have been added at lines 175–180 on page 7 as follow: "For further analysis of the modulation of the co-occurrence of $O_3$–$PM_{2.5}$ pollution by the boundary layer structure in some local areas, we also used the BLH, uv-wind, vertical velocity, RH and temperature fields of the fifth generation European Centre for Medium-Range Weather Forecasts reanalysis (ERA5), which has a high spatiotemporal resolution (0.25° × 0.25°, hourly; https://cds.climate.copernicus.eu/cdsapp#!/home)."

Our original intention was to explore the regulation of large-scale synoptic circulation on compound pollution. Guan and Li (2021) proposed that the correlation coefficient of geopotential height between NCEP reanalysis data and scientific experimental data (the third Qinghai–Tibet Plateau atmospheric science experiment from 2015 to 2017) is above 0.99. Consequently, we selected geopotential heights from NCEP data as a categorical variable. For further analysis of the modulation of the co-occurrence of $O_3$–$PM_{2.5}$ pollution by the boundary layer structure, we used ERA5 data (such as hourly BLH, temperature data, etc.) with a high spatiotemporal resolution as well. In order to strengthen the robustness of our work, we provided a figure of four SWPs based on ERA5 reanalysis data (Fig. S2), which is highly consistent with NCEP reanalysis data at large scales (Figs. 4 and S2). Additionally, we also furtherly compared the differences between NCEP and ERA5 data. As shown in Fig. S3, the geopotential height of NCEP reanalysis data is significantly positively correlated with that of ERA5 data. Especially in eastern China, the correlation

coefficient between the two is greater than 0.96, and all of our classification areas have passed the 99% level of significance test. Overall, the results of this study are robust. We have inquired into the influence of local boundary layer structure on compound pollution events based on the hourly PBL and other meteorological variables of ERA5 data. This has deepened our understanding of the mechanism of the compound pollution events in eastern China during summertime.

[Figure]

Fig. S2. As in Fig. 4 but for ERA5 reanalysis data.

[Figure]

**Fig. S3. The correlation of geopotential height between NCEP and ERA5 reanalysis data. The shading indicates the correlation, and the black dots indicate passing the 99% level of significance test.**

Reference:

Guan, Q., Li, Q., SHI, C., HU, Y., MEI, C. and ZHANG, N.: Evaluation of Reanalysis Data Based on the Three-dimensional High-density Sounding Data of the Qinghai-Tibet Plateau, Meteorol. Environ. Res., 12(1), 34-41+51, doi:10.19547/j.issn2152-3940.2021.01.007, 2021.

4. How many sounding profiles at 08, 14 and 20 BJT were used in this study for each studied city? How to use 08 and 20 LT soundings to estimate the afternoon BLH? Please clarify. In summer, the relationships between BLH and concurring/compound pollution in East China are quite complicated due to the transport of precursors (https://doi.org/10.1016/j.envpol.2020.115775). More in-depth analysis/discussion on the PBL-pollution linkage and transport of precursors in East China must be added.

**RESPONSE: Thanks for your constructive suggestions. We have clarified this at lines 162–166 on pages 6-7 as follow: "Surface meteorological data, such as Tmax, precipitation, WS and RH from 611 meteorological observation stations, along with sounding data at 1400 Beijing time (BJT) from 64 stations and at 0800 BJT and 2000 BJT from 77 stations, in eastern China, were obtained from the China National Meteorological Information Center of the China Meteorological Administration (http://data.cma.cn/site/index.html).". The transportation of precursors and PBL-pollution linkage have been added in Discussion. Please also see as follow:**

**"In particular, Type 1 had significantly warmer temperatures over the boundary layer during the compound pollution periods of the BTH region, as compared with the clean periods. The daytime BLH under the compound pollution condition was also higher than that under the clean condition. In addition, there were different directions of prevailing winds during the two periods. The prevailing southerly winds during the compound pollution period may have driven the transportation of air pollutants from the southern plains, resulting in more serious pollution (Fig. 11; see also. Miao et al. (2020) also proposed another mechanism—that is, the synoptic southerly warm advections at the top of PBL, can strengthen the elevated thermal inversion layer and suppress the development of the PBL, causing worse pollution. Co-influenced by the topographical effect of the northern mountainous areas and the boundary layer structure, air pollutants could be trapped in the BTH region. In comparison, although there was a southerly prevailing wind in the BTH region (Figs. 11 and S14), the rain belt also being located in the southern area of the BTH might have led to the potential removal of $PM_{2.5}$ (Fig. 9j). Therefore, compound pollution across the BTH region might mainly have been due to local emissions of air pollutants".**

5. Please carefully check the cited papers, some were not properly. For example, the BLH estimation method was actually from the study of Seidel et al. (2012, https://doi.org/10.1029/2012JD018143).
**RESPONSE: Sorry for our confusion. We have carefully checked the cited papers and revised in the latest manuscript. Thanks for your attention.**

**Anonymous Referee #2:**
The authors use the T-mode PCA to objectively classify the summertime synoptic weather pattern across East-Asia and the western Pacific Basin aiming to identify the mode(s) most favorable for compound pollution events across sub-regions in China, specifically for PM2.5 and O3. Many factors governing these events operating across an array of scales are explored. The PCA identified 4 synoptic regimes characterizing the seasonal set up of the 500 hPa WPSH from 2015-2018. An additional large-scale circulation is also at work here, the East-Asian monsoon, which is discussed in context to the WPSH. Additionally, the authors discuss the effects of precipitation frequency and boundary layer characteristics on regulating compound pollution events. Occurrences of pollution are based on Chinese governmental standards.
The authors present a much-improved manuscript. The authors now show a clear connection between different synoptic modes and compound pollution events across different sub-regions in eastern Asia. The authors link the favorable synoptic modes to favorably meteorological conditions in Tmax, wind, stability, and more.
I believe that this paper will be ready for publication once its grammar has been improved. Thus, I recommend major revisions at this time, but I must emphasize that the authors should be proud of the improvements they have made to this manuscript. There is a strong message developing. With the proper grammatical improvements, this will be a significant contribution to the literature.
**RESPONSE: Thank you very much for your high recognition of our work, and we believe that our revised manuscript will be further improved under your constructive suggestions. Please find our point-by-point responses below.**
1. The abstract can and should be shortened considerably. The authors have identified two

preferred SWPs conducive for compound pollution events and then provide many details. The details can be left to the main text and omitted from the abstract. Furthermore, the abstract should be in the same tense. Currently, there is a mix of past tense and present tense expository.

**RESPONSE: Thanks for your constructive suggestion. We have shortened the abstract and carefully checked the sentence tense, please also see as follow: "Surface ozone ($O_3$) pollution during summer (June–August) over eastern China has become more severe in recent years, resulting in a co-occurrence of surface $O_3$ and $PM_{2.5}$ (particulate matter with aerodynamic diameter $\leq$ 2.5 μm in the air) pollution. However, the mechanisms regarding how the synoptic circulation pattern might influence this compound pollution remain unclear. In this study, we applied the T-mode principal component analysis (T-PCA) method to objectively classify the occurrence of four synoptic weather patterns (SWPs) over eastern China, based on the geopotential heights at 500 hPa during summer (2015–2018). These four SWPs over eastern China were closely related to the western Pacific subtropical high (WPSH), exhibiting significant intraseasonal and interannual variations. Based on ground-level air quality observations, remarkable spatial and temporal disparities of surface $O_3$ and $PM_{2.5}$ pollution were also found under the four SWPs. In particular, there were two SWPs that were sensitive to compound pollution (Type 1 and Type 2). Type 1 was characterized by a stable WPSH ridge with its axis at about 22°N and the rain belt located in the south of the Yangtze River Delta (YRD); and Type 2 also exhibited WPSH dominance (ridge axis at ~25°N), but with the rain belt (over the YRD) at a higher latitude compared to Type 1. In general, SWPs have played an important role as driving factors of surface $O_3$–$PM_{2.5}$ compound pollution in a regional context. Our findings demonstrate the important role played by SWPs in driving regional surface $O_3$–$PM_{2.5}$ compound pollution, in addition to the large quantities of emissions, and may also provide insights into the regional co-occurring high levels of both $PM_{2.5}$ and $O_3$ via the effects of certain meteorological factors."**

2. Line 79: What does "gradually been prominent" mean? Do the authors mean that O3 pollution in summer has increased in recent years?

**RESPONSE: Indeed, the $O_3$ pollution in summer has increased in recent years. "For instance, Sun et al. (2016) showed that the observed summertime $O_3$ at Mt. Tai increased significantly by 1.7 ppbv $yr^{-1}$ for the month of June and 2.1 ppbv $yr^{-1}$ for the months of July–August during the period of 2003 to 2015. Furthermore, an increase in the maximum daily 8-h average concentration of $O_3$ (MDA8 $O_3$) at an annual-average rate of 4.6%, was reported by Fan et al. (2020), albeit with a decrease in the frequency of $PM_{2.5}$ pollution." are shown at lines 72–76 on pages 3.**

References:

Fan, H., Zhao, C. and Yang, Y.: A comprehensive analysis of the spatio-temporal variation of urban air pollution in China during 2014–2018, Atmos. Environ., 220(November), 117066, doi:10.1016/j.atmosenv.2019.117066, 2020.

Sun, L., Xue, L., Wang, T., Gao, J., Ding, A., Cooper, O. R., Lin, M., Xu, P., Wang, Z., Wang, X., Wen, L., Zhu, Y., Chen, T., Yang, L., Wang, Y., Chen, J. and Wang, W.: Significant increase of summertime ozone at Mount Tai in Central Eastern China, Atmos. Chem. Phys., 16(16), 10637–10650, doi:10.5194/acp-16-10637-2016, 2016.

3. Line 90: WS, please define as wind speed. I do not it is defined previously in the main text.

**RESPONSE: Sorry for our negligence, and thank you for your reminder. We have defined "WS" as wind speed at line 80 on page 4.**

4. Line 105: When referring to previous studies, present material in the past tense, but the rest of the paper should be written in the present tense.
**RESPONSE: Thanks for your kind suggestion. We have revised this sentence to "Miao et al. (2015) showed that RH was high when aerosol pollution occurred in the BTH region."**

5. Line 116: "anomalies" should be "anomaly".
**RESPONSE: Thanks, and revised.**

6. Line 129: Are the winds southerly or northwesterly?
**RESPONSE: The weak northwesterly prevailing winds was related to local emissions of aerosols, while the southerly prevailing winds was related to the transportation of pollutants from southern cities to Beijing.**

7. Line 137: Delete "simulation"
**RESPONSE: Deleted, and thanks.**

8. Line 146: Should be "pollutants"
**RESPONSE: Thanks, and revised.**

9. Line 192: "consists" should be "consisting"
**RESPONSE: Thanks, and revised.**

10. Line 193: "pattern" should be "patterns"
**RESPONSE: Thanks, and revised.**

11. Lines 194-197: This sentence needs to be reworked grammar-wise.
**RESPONSE: Thank you for your kind suggestion. We have revised it as follow: "First, the weather data are spatially standardized and split into 10 subsets by T-PCA. Then the principal components (PCs) of weather information are estimated by applying singular value decomposition, and the PC score for each subset can be calculated after oblique rotation. Finally, the resultant subset with the highest sum will be selected by comparing 10 subsets according to contingency tables, and its types can be output as well (Miao et al., 2017; Philipp et al., 2014).".**

12. Line 205: Are the authors counting days as O3 and PM2.5 days when > 50% of the sites exceed the aforementioned thresholds? If so, the grammar here needs to be reworked.
**RESPONSE: Thanks for your advice. We have rewritten this sentence as "In this study, we characterized regional pollution days as occurring when the average values of more than 50% of sites in this region exceeded the aforementioned thresholds.".**

13. Line 231: "The" should be "the"

**RESPONSE: Thanks, and revised.**

14. Line 232: New sentence should begin at ", as a result"
**RESPONSE: Thanks for your comments, and revised.**

15. Lines 233-237: Are the authors referring to the total days in the 2015-2018 period?
**RESPONSE: Sorry for our unclear statement. The results refer to the summer of 2015-2018. We have revised it as follow: "During the study period, the number of days of O3 pollution in the BTH, YRD, PRD, GZP, and NEM regions was 254, 133, 84, 165 and 96 respectively, while the number of days of PM2.5 pollution was only 93, 8, 0, 2 and 1, of which compound pollution occurred on 76, 7, 0, 2, and 0 days according to Chinese standards (the asterisks in Fig. 3 indicate the compound pollution events)."**

16. Line 237: This sentence is repeated
**RESPONSE: Thanks for your kind reminder. We have deleted this sentence in the revised manuscript.**

17. Line 255: Wait – are Figs. 2-3 composited only on days characterized by SWPs 1-4? I thought these for all days? If for all days, delete "days for four SWPs"
**RESPONSE: If I am not mistaken, it is line 245? Yes, all days during the research period was classified into SWPs 1-4. Thank you for your scrupulous review and sorry for our carelessness. This sentence should be "Based on this target, the number of pollution days for the five urban clusters were 194, 52, 16, 47, and 20, respectively (Fig. 3).".**

18. Lines 247-248: Change the wording of this sentence." These results indicate that, despite PM2.5 reductions, compound pollution events deserve public attention." Delete the following sentence.
**RESPONSE: Thanks for your constructive comment. And revised.**

19. Lines 261-263: This sentence needs to be reworded from "which might…" onwards
**RESPONSE: Thanks for your constructive comment. We have reworded this sentence as follow: "Low-level southerly monsoonal flow forming at the periphery of an anomalously enhanced WPSH, along with the transportation of warm and humid air from the ocean to East Asia, might also be responsible for the asymmetric spatial distribution of ground-level O₃ [i.e., a decrease in southern China but an increase over northern China (Zhao & Wang, 2017)]".**

20. Line 273: Change "in" to "across"
**RESPONSE: Thanks, and revised.**

21. Lines 275-278: This sentence is hard to follow and needs to be reworked. For example, how can the sea-land interaction interact with the southeastern region across China? I think the authors can just explain the different spatial configurations of the different modes of the WPSH and leave discussion for later on when discussing the compound pollution event conditions.

**RESPONSE: Thanks for your constructive comment. We have rewritten this sentence as "The southwest wind from the South China Sea might have combined with the southerly wind in the eastern periphery of the WPSH. As a result, southerly winds prevailed across southeastern China, while northern China was mainly controlled by the westerly trough.".**

22. Line 535: "locating" should be located
**RESPONSE: If I am not mistaken, it is line 355? Thanks, and revised.**

23. Line 415: Prevailing…." winds?"
**RESPONSE: Thanks for your kind reminder. We have added "winds" to after "prevailing".**

24. Lines 418-420: Were the prevailing winds driving pollution transport from the southern plains? This sentence needs to be reworked grammatically.
**RESPONSE: We appreciate your suggestions. Yes, they were. We have reworded it as "In addition, there were different directions of prevailing winds during the two periods. The prevailing southerly winds during the compound pollution period may have driven the transportation of air pollutants from the southern plains, resulting in more serious pollution (Fig. 11; see also Miao et al., 2019, 2020).".**

25. Fig. 11: Panels are uneven. Please replot
**RESPONSE: Thanks, and replotted.**

[Figure]

**Fig. 11. Daily variations of horizonal wind, potential temperature and BLH in the BTH area during clean and compound pollution periods under Type 1 and Type 2 (a, b, e, f). The vertical cross-section of u-wind, w-wind and potential temperature for the same situation in the BTH region (c, d, g, h). The w-wind is multiplied by 100 when used. The data are from the ERA5 reanalysis.**

26. Lines 427-463: These points can be shortened, and the grammar needs to be revised. A lot of

the discussion for this passage was made in previous sections.

RESPONSE: Thanks for your constructive suggestion. We have shortened and revised these points. Please also see as follow:

"(1) Type 1: Under the conditions of high temperatures (Tmax > 27°C), moderate humidity (RH ~60%), and low PF, photochemical reactions were greatly promoted to cause severe $O_3$ pollution. Meanwhile, the BTH–NYRD areas were located in front of the westerly trough, under the influence of the warm and humid air of the WPSH, and so the hygroscopic growth of fine particulates potentially caused a certain amount of $PM_{2.5}$ pollution (Li et al., 2017; Zhang et al., 2016b), becoming $O_3$–$PM_{2.5}$ compound pollution (Fig. 12). In addition, the prevailing southerly winds in the boundary layer were able to transport the pollutants emitted from southern cities to the BTH, atmospheric stratification was stable when the air mass was sinking (Miao et al., 2019b; Figs. 11 and S12), and compound pollution may have been especially severe. Although a relatively higher BLH occurred in the BTH region, the prevailing southerly winds in the boundary layer served to further increase the pollution.

(2) Type 2: $O_3$ pollution was severe under the meteorological conditions of high temperatures, moderate humidity, and weak precipitations. The $PM_{2.5}$ in the BTH region, which was located in front of the westerly trough, was high since the shallow boundary layer and low wind frequency were unfavorable for the diffusion of pollutants. Therefore, $O_3$–$PM_{2.5}$ compound pollution was also rather frequent (Fig. 12).

(3) Type 3: High temperatures, low humidity, and weak precipitations over the YRD region tended to generate a large amount of $O_3$, while the positive BLH and negative FLWD anomalies were unfavorable to $O_3$ accumulation. On the other hand, summer typhoon activities might have weakened the WPSH intensity over the YRD region, leading to the eastward retreat and northward shift of the WPSH. As a result, the high WS across coastal areas was able to ease the ground-level $O_3$ pollution (Shu et al., 2016). For the BTH and PRD regions, the high PF tended to suppress the production of $O_3$.

(4) Type 4: High temperatures, medium-high humidity and weak precipitations in the GZP and PRD regions were able to cause $O_3$–$PM_{2.5}$ compound pollution, but the $PM_{2.5}$ pollution in both regions was not heavy, possibly in relation to local lower emissions of pollutants. Under the control of the WPSH, there were strong photochemical reactions at high temperatures and little rainfall in some eastern regions (such as the northern BTH, YRD), which was also conducive to $O_3$ generation (Fig. 12). Meanwhile, relative to Type 1, $O_3$ pollution was lighter in the BTH, due to the differences of RH, BLH and FLWD.".

---

## Author Response (AR3)

**Reponses to referee(s) comments**

**Dear Editor,**

**Thank you for your efforts for handling our manuscript. We appreciate to receive the useful comments from reviewer. These comments are very constructive, and we have now further revised our manuscript in light of referee's comments. Based on the helpful suggestions from reviewer, we believe that we should have addressed questions and concerns from referee appropriately, and adequately. Please find our point-by-point responses below.**

Suggestions for revision or reasons for rejection (will be published if the paper is accepted for final publication)

The authors have addressed most of my concerns. The manuscript can be accepted after minor revision.

**RESPONSE: Thanks for your constructive comments. We have revised all your problems carefully. Please find our point-by-point responses below.**

(1) How many sounding profiles (the exact number) at 08:00, 14:00, and 20:00 BJT were used in this study?

**RESPONSE: Thanks for your suggestion. Sorry for missing the number of sounding profiles. 368,367, and 368 sounding profiles at 0800, 1400, and 2000 BJT were used in this study. We have clarified this issue at lines 164-165 on page 7 as "along with 367 sounding profiles at 1400 Beijing time (BJT) from 64 stations and 368/368 sounding profiles at 0800/2000 BJT from 77 stations, respectively".**

(2) In Fig. 11, clear differences can be observed in the same type. Were the clean samples relevant to the precipitation processes? The variations of clean and pollution days may be primarily caused by the precipitation, not the PBL structure. Please clarify.

**RESPONSE: Many thanks for your constructive suggestion. We have clarified it at lines 428-429 on page 16. Please see also as follow: "It can be clearly seen that various precipitation primarily caused differences in concentrations of both $O_3$ and $PM_{2.5}$ between clean and pollution days under Type 1/Type 2 (See Figs. 12–13).".**

[Figure]

**Fig. 12. Precipitation, WS, and WD during clean and compound pollution periods under Type 1 over BTH.**

[Figure]

**Fig. 13. As in Fig. 12 but for Type 2.**

(3) Line 438, "The prevailing southerly winds during the compound pollution period may have driven the transportation of air pollutants from the southern plains, resulting in more serious pollution (Fig. 11; see also." The sentence is incomplete, please check.

**RESPONSE: Thanks for your suggestion. We have completed this sentence as "The prevailing southerly winds during the compound pollution period may have driven the transportation of air pollutants from the southern NCP, resulting in more serious pollution (Fig. 11), which is consistent with the results of Miao et al. (2017, 2019).".**

(4) Line 443-446, "In comparison, although there was a southerly prevailing wind in the BTH region (Figs. 11 and S14), the rain belt also being located in the southern area of the BTH might have led to the potential removal of PM2.5 (Fig. 9j). Therefore, compound pollution across the BTH region might mainly have been due to local emissions of air pollutants." The rain belt is not always fixed in the southern area of BTH, which cannot support the conclusion "pollution across the BTH region might mainly have been due to local emissions of air pollutants".

**RESPONSE: Many Thank you for your valuable comments. We have restated our points as following: "In comparison, although there was a southerly prevailing wind in the BTH region under Type 2 (Figs. 11 and 13), the rain belt being located in the southern area of the BTH might have led to the potential removal of $PM_{2.5}$ over there (Fig. 9j), so the pollutants transported from the southern NCP would be partially reduced. Therefore, it can be concluded that the emissions of local pollutants accompanied with unfavorable meteorological conditions will continuously accumulate pollutants (Figs. 8–9 and 12–13; Gui et al., 2019; Zhang et al., 2020), which should be main cause of the BTH compound pollution."**

---

## Author Response (AR4)

Comments to the Author:
Dear Authors,

Your manuscript has been much improved after revisions. I am accepting the manuscript for publication in ACP. However, before uploading your manuscript, please have a professional English language service or a native English speaker to read through the manuscript to improve the grammar of your manuscript.

Best regards,
Xiaohong Liu

**Responses to the Editor:**
**Dear Editor,**
**Thank you for your efforts for handling our manuscript. We have employed an English-language editing service to polish our manuscript. Certification attached as following.**

**Certificate**

[Figure]

| | |
|---|---|
| **Reference number**: 2021-YangYuanjian-7-R1 | **Date: 12 May 2021** |
| **Contact author:** Yuanjian Yang | **Manuscript:** Large-scale synoptic drivers of co-occurring summertime ozone and PM2.5 pollution in eastern China |

This document certifies that the above-detailed manuscript was edited by a native English-speaking expert at LucidPapers on the date stated.

Following the editing process, the editor's overall assessment is that:

| | |
|---|---|
| The manuscript will be ready for consideration by the target journal once the edits have been checked and approved/rejected as necessary. | |
| The manuscript may require modifications to the text in response to the editor's changes and comments/queries, but a second check of any such modifications is unlikely to be needed before sending to the target journal. | ☑ |
| The manuscript requires modifications to the text in response to the editor's changes and comments/queries, and a second check of any such modifications might be advisable before sending to the target journal. | |
| The manuscript requires modifications to the text in response to the editor's changes and comments/queries, and a second check of these changes is recommended before sending to the target journal. | |
| The manuscript requires major changes, rewriting and restructuring and a second edit of the entire paper is strongly recommended before sending to the target journal. | |

Signed:

**Colin Smith**
**Chief Editor**
**LucidPapers**

Email: colin.smith@lucidpapers.com
Website: http://www.lucidpapers.com